# R2PS: Worst-Case Robust Real-Time Pursuit Strategies under Partial Observability

**Runyu Lu**[1,2]**, Ruochuan Shi**[2,1]**, Yuanheng Zhu**[2,1,†]**, Dongbin Zhao**[2,1]

[1] School of Artificial Intelligence, University of Chinese Academy of Sciences[*]
[2] State Key Laboratory of Multimodal Artificial Intelligence Systems,
Institute of Automation, Chinese Academy of Sciences
`lurunyu17@mails.ucas.ac.cn`
`{ruochuan.shi,yuanheng.zhu,dongbin.zhao}@ia.ac.cn`

## Abstract

Computing worst-case robust strategies in pursuit-evasion games (PEGs) is time-consuming, especially when real-world factors like partial observability are considered. While important for general security purposes, real-time applicable pursuit strategies for graph-based PEGs are currently missing when the pursuers only have imperfect information about the evader's position. Although state-of-the-art reinforcement learning (RL) methods like Equilibrium Policy Generalization (EPG) and Grasper provide guidelines for learning graph neural network (GNN) policies robust to different game dynamics, they are restricted to the scenario of perfect information and do not take into account the possible case where the evader can predict the pursuers' actions. This paper introduces the first approach to worst-case robust real-time pursuit strategies (R2PS) under partial observability. We first prove that a traditional dynamic programming (DP) algorithm for solving Markov PEGs maintains optimality under the asynchronous moves by the evader. Then, we propose a belief preservation mechanism about the evader's possible positions, extending the DP pursuit strategies to a partially observable setting. Finally, we embed the belief preservation into the state-of-the-art EPG framework to finish our R2PS learning scheme, which leads to a real-time pursuer policy through cross-graph reinforcement learning against the asynchronous-move DP evasion strategies. After reinforcement learning, our policy achieves robust zero-shot generalization to unseen real-world graph structures and consistently outperforms the policy directly trained on the test graphs by the existing game RL approach.

## 1 Introduction

Pursuit-evasion game (PEG) is an important topic long examined in the fields of robotics and security (Vidal et al., 2001; 2002; Chung et al., 2011). Many real-world tasks can benefit from the solution to an abstracted PEG, e.g., guiding a team of cops to capture a robber and aligning a team of guards to defend against an intruder. In comparison with traditional differential games (Margellos & Lygeros, 2011; Zhou et al., 2012), graph-based PEGs are convenient for describing complicated scenarios, possibly with a large scale. When we use graphs as a common structural representation, the actions of the pursuers and the evader can be abstracted as moving from a vertex to an adjacent one at each discrete timestep. The edges between the vertices can possibly represent urban streets in reality.

However, exactly solving graph-based PEGs is computationally expensive (see Goldstein & Reingold (1995)). Even under a slight structural change, the worst-case robust pursuit strategies can be different and thus require a large amount of time to be recomputed. For example, when a traffic jam happens in the city, the related edges in the PEG graph can be frequently removed and added. This severely limits the real-time applicability of the existing methods featuring mathematical programming (Vieira et al., 2008; Horák & Bošanský, 2017). Besides, real-world factors like partial observability, which

---

[*]This work was supported in part by the National Natural Science Foundation of China under Grants 62293541 and 62136008 and in part by the Beijing Nova Program under Grant 20240484514.

leads to PSPACE-hardness even under a fixed opponent (see Papadimitriou & Tsitsiklis (1987)), further increase the difficulty of deriving a well-performing pursuit strategy within a time limit.

Reinforcement learning (RL), which has demonstrated strong generalization capabilities in domains like large language models (see Chu et al. (2025)), provides an alternative solution to this problem. We may train a parameterized policy represented by a suitable neural network, e.g., a graph neural network (GNN) (Wu et al., 2020), on a diverse set of graphs and then generalize it to the unseen graph structures. Unfortunately, while RL has been applied to solving large-scale PEGs (Xue et al., 2022; 2021), existing research focuses more on its scalability rather than generalization capability. The methods like MT-PSRO (Li et al., 2023) and Grasper (Li et al., 2024) are limited to few-shot generalization to unseen opponent strategies and initial conditions. As is pointed out by Zhuang et al. (2025), they still have difficulty adapting to rapid changes of graph structures. The state-of-the-art method, Equilibrium Policy Generalization (EPG) (Lu et al., 2025a), first examines zero-shot generalization at the level of graphs. However, whether the paradigm of EPG works under partial observability remains underexplored. Besides, all of the mentioned works do not consider the possible case that the evader may have stronger observation capabilities than the pursuers. This makes the strength of the learned pursuit strategies less convincing for real-world security purposes.

In this paper, we present an approach to finding pursuit strategies that are both worst-case robust and real-time applicable under partial observability. We start by analyzing a dynamic programming (DP) algorithm for efficiently solving Markov PEGs and proving that it also finds optimal strategies when the evader can predict the pursuer's action and move asynchronously. With a belief update mechanism, we further extend the DP policies to a partially observable setting. The belief preservation serves to avoid the complexity of recording all observation histories through abstracting opponent information for effective decision-making. Finally, we embed the belief preservation mechanism into the reinforcement learning framework of EPG and train a generalized GNN pursuer policy under partial observability. We then evaluate the worst-case robustness of our real-time RL pursuer policy.

Specifically, the contributions of this paper are threefold:

- We theoretically analyze a dynamic programming (DP) algorithm and extend the optimal strategies induced by this algorithm to asynchronous-move and partially observable scenarios. We prove that the DP algorithm induces strictly optimal pursuit and evasion strategies when the evader moves asynchronously and design a belief preservation mechanism against the possibly unobserved evaders. Under belief preservation, we verify that the extended pursuer policy remains strong against the provably optimal perfect-information evader.

- We practically train an observation-based pursuer policy across different graph structures, deriving the first worst-case robust real-time pursuit strategies (R2PS) applicable to dynamically changing PEGs with partial observations. We combine our belief preservation mechanism with the state-of-the-art robust policy generalization paradigm, EPG, and provide an inference time complexity bound for our GNN-represented RL pursuer policy.

- Through extensive experiments, we verify that under partial observability, our RL training against the asynchronous-move DP evaders under a diverse set of graphs leads to robust zero-shot performance in unseen real-world graphs. Comparative results reveal the superiority of our RL approach over the standard game RL approach, PSRO (Lanctot et al., 2017).

## 2 PRELIMINARIES

### 2.1 PROBLEM FORMULATION

Adversarial games with partial observability can be generally represented by partially observable stochastic games (POSGs), where equilibrium learning has been rigorously examined in existing game-theoretic research (e.g., Lu et al. (2025b;c)). However, this formulation considers all possible observation histories and leads to a large set of decision points whose size is possibly exponential in the time horizon of the game. For the worst scenario of pursuit-evasion, while the pursuers have limited observation capabilities, the evader could still obtain the global information of the game. Since at least one side of the players possesses perfect information, we consider first expressing PEGs as two-player zero-sum Markov games and then extending the definitions to incorporate practical adversarial factors like partial observability and asynchronous moves of the evader.

**Two-player zero-sum Markov game.** An infinite-horizon two-player zero-sum Markov game is represented by a tuple $(S, \mathcal{A}, \mathcal{B}, \mathcal{P}, r, \gamma)$, where $S$ is the state space, $\mathcal{A}$ is the action space of the max-player (who aims to maximize the cumulative reward), $\mathcal{B}$ is the action space of the min-player (who aims to minimize the cumulative reward), $\mathcal{P} \in [0,1]^{|S||\mathcal{A}||\mathcal{B}| \times |S|}$ is the transition probability matrix, $r \in [0,1]^{|S||\mathcal{A}||\mathcal{B}|}$ is the reward vector, and $\gamma \in (0,1)$ is the discount factor. In PEGs, the max-player is the team of $m$ pursuers, and the min-player is the evader. We use a termination function $f : S \to \{0,1\}$ to mark the states where the pursuit is successful. When $f(s) = 1$, the game is terminated, and a reward of $+1$ is received. Otherwise, a reward of $0$ is received. The discount factor $\gamma < 1$ encourages the pursuers to capture the evader as soon as possible.

**Graph-based pursuit-evasion game.** Considering the requirements of formulating large-scale real-world scenarios, we describe states and actions on a graph structure $G = \langle \mathcal{V}, E \rangle$: $\mathcal{V}$ is the set of vertices $v$. The global state $s = (s_p, s_e)$ in a game is an element of $\mathcal{V}^m \times \mathcal{V}$, where $s_p = (v_p^1, v_p^2, \cdots, v_p^m) \in \mathcal{V}^m$, and $s_e = v_e \in \mathcal{V}$. An edge $e = (v, v') \in E$ defines the adjacency between two vertices $v, v' \in \mathcal{V}$. For example, when we represent an urban scenario by a graph $G$, an edge $e$ can be used to describe a unit length of streets. The valid actions of the $m + 1$ agents in a graph-based PEG are either moving to an adjacent vertex via an edge or staying at the current node.

**Policy and value function.** Following common notations, we denote by $(\mu, \nu)$ the joint policy of the two players, where $\mu$ is the policy of the max-player (pursuers) and $\nu$ is the policy of the min-player (evader): $\mu(s) \in \Delta(\mathcal{A})$ (resp., $\nu(s) \in \Delta(\mathcal{B})$) is the max-player's (resp., min-player's) action distribution at state $s \in S$. Since $\Delta(\mathcal{A})$ is the probability simplex over $\mathcal{A}$, $\mu(s, a)$ corresponds to the probability of selecting action $a \in \mathcal{A}$ at state $s$. Given the joint policy, we further define the value function $V^{\mu,\nu}(s) = \mathbb{E}\left[\sum_{t=0}^{\infty} \gamma^t r(s_t, a_t, b_t) \,|\, s_0 = s; \mu, \nu\right]$ as in Markov decision processes.

**Solution concept.** A Nash equilibrium (NE) in a game is a joint policy where each individual player cannot benefit from unilaterally deviating from his/her own policy (Roughgarden, 2016). Specifically, in a two-player zero-sum MG, an NE $(\mu^*, \nu^*)$ satisfies $V^{\mu,\nu^*} \leq V^{\mu^*,\nu^*} \leq V^{\mu^*,\nu}$ for any $\mu$ and $\nu$ at all states. As is well known, every MG with finite states and actions has at least one NE, and all NEs in a two-player zero-sum MG share the same value $V^*(s) = V^{\mu^*,\nu^*}(s) = \max_\mu \min_\nu V^{\mu,\nu}(s) = \min_\nu \max_\mu V^{\mu,\nu}(s)$ (Shapley, 1953). In two-player zero-sum Markov games, Nash equilibrium can be viewed as a globally optimal joint policy since both players cannot be exploited by their worst-case opponents when the players move synchronously (simultaneously).

**Game extension.** Since Markov games only take into account synchronous moves and full observations, we further allow for two variations concerning asynchronous moves and partial observability. In reality, the worst evader (from the pursuers' perspective) may have good predictions of the pursuit actions. Therefore, we allow it to decide after the pursuers' move $a$ at each timestep. In this case, the evader policy $\nu(s)$ is transformed into an asynchronous one $\nu(s, a)$, and we say that a strategy is optimal for the pursuer/evader side at state $s$ if the worst-case termination timesteps of all possible trajectories starting from $s$ are maximized/minimized. Besides, the availability of sensors may not allow the pursuers to observe an agent that is far away (while the worst evader can). In this case, the pursuer policy $\mu(s)$ is transformed into $\mu(o)$, where $o$ is the history of the pursuers' local observations.

## 2.2 DYNAMIC PROGRAMMING FOR MARKOV PEGS

The traditional marking algorithm (Chung et al., 2011) provides a general idea of recursively finding optimal strategies in perfect-information PEGs. If all possible evading actions lead to the states that have been marked, then we can also mark the current state, which means the pursuers can capture the evader starting from this state. However, a direct implementation of the marking algorithm incurs a time complexity much higher than the theoretical lower bound $\Omega(|S|)$. In view of this gap, Lu et al. (2025a) introduce a dynamic programming (DP) algorithm (see Algorithm 1) that guarantees near-optimal time complexity for solving Markov PEGs.

Algorithm 1 computes a distance table $D$ through preserving a queue $\mathcal{Q}$. Intuitively, the distance value $D(s)$ indicates the worst-case timestep for the pursuer side to capture the evader starting from the global state $s = (s_p, s_e)$, which is guaranteed through the use of a minimax policy

$$\mu^*(s_p, s_e) = \operatorname*{arg\,min}_{\text{neighbor } n_p \text{ of } s_p} \left\{ \max_{\text{neighbor } n_e \text{ of } s_e} D(n_p, n_e) \right\}. \tag{1}$$

---

**Algorithm 1:** Dynamic Programming for Markov PEGs

---

**Input:** Graph $G = \langle \mathcal{V}, E \rangle$, Pursuer Number $m$, and Termination Function $f : \mathcal{V}^m \times \mathcal{V} \to \{0, 1\}$

1   Initialize an empty queue $\mathcal{Q}$ and the distance table $D = \infty$
2   **for** *pursuer state (positions) $s_p \in \mathcal{V}^m$* **do**
3     **for** *evader state $s_e \in \mathcal{V}$* **do**
4       **if** $f(s_p, s_e) = 1$ **then**
5         $D(s_p, s_e) \leftarrow 0$
6         Push $(s_p, s_e)$ into $\mathcal{Q}$
7       **end**
8     **end**
9   **end**
10   **while** *$\mathcal{Q}$ is not empty* **do**
11     Pop the first element $(s_p, s_e)$ from $\mathcal{Q}$
12     **for** *evader neighbor $n_e \in$ Neighbor$(s_e)$, $\nexists n'_e \in \mathcal{V}, (n_e, n'_e) \in E, D(s_p, n'_e) > D(s_p, s_e)$* **do**
13       **for** *pursuer neighbor $n_p \in$ Neighbor$(s_p) \subset \mathcal{V}^m, D(n_p, n_e) = \infty$* **do**
14         $D(n_p, n_e) \leftarrow D(s_p, s_e) + 1$
15         Push $(n_p, n_e)$ into $\mathcal{Q}$
16       **end**
17     **end**
18   **end**

**Output:** Distance Table $D$

---

Under synchronous moves, the evader's policy is symmetrically defined as

$$\nu^*(s_p, s_e) = \underset{\text{neighbor } n_e \text{ of } s_e}{\arg\max} \left\{ \min_{\text{neighbor } n_p \text{ of } s_p} D(n_p, n_e) \right\}. \tag{2}$$

Using mathematical induction, Lu et al. (2025a) prove that the joint policy $(\mu^*, \nu^*)$ is a near-optimal pure strategy (the proof can be found in Appendix A.1):

**Theorem 1.** *If there exists a pure-strategy Nash equilibrium in the Markov PEG, then the joint policy $(\mu^*, \nu^*)$ defined by (1) and (2) is a Nash equilibrium.*

## 3   EXTENDING DYNAMIC PROGRAMMING POLICIES TO ASYNCHRONOUS MOVES AND PARTIAL OBSERVABILITY

In this section, we further show that the distance table $D$ generated by the DP algorithm (Algorithm 1) can also be used to construct the optimal evader policy under asynchronous moves, as well as the observation-based pursuer policies under partial observability.

### 3.1   ASYNCHRONOUS-MOVE SETTING

When the evader moves asynchronously, we define the DP policy for the evader as

$$\nu^*(s_p, s_e, n_p) = \underset{\text{neighbor } n_e \text{ of } s_e}{\arg\max} \{D(n_p, n_e)\}, \tag{3}$$

where $n_p$ is the neighbor of $s_p$ that the pursuers choose to move to in the current decision step, which is perceived or predicted by the evader in advance. With this information as an additional input, the evader can decide based on the pursuers' positions after their decision rather than before. As a result, the policy (3) no longer requires the inner enumeration in (2).

In this case, we can show that the pursuer policy (1) and evader policy (3) induced by the distance table $D$ are strictly optimal at all states. We start our analysis by proving Lemma 1, which reveals the minimax essence of the distance table $D$. The detailed proof can be found in Appendix A.2.

**Lemma 1.** *When $D(n_p, n_e) > 0$, Algorithm 1 guarantees that*

$$D(n_p, n_e) = \min_{\text{neighbor } s_p \text{ of } n_p} \left\{ \max_{\text{neighbor } s_e \text{ of } n_e} D(s_p, s_e) \right\} + 1.$$

Using Lemma 1, we can further prove that $D(s)$ implies the best possible worst-case timesteps starting from state $s$ for both pursuer and evader sides under the asynchronous-move setting. The main results are shown as follows, and the omitted proofs can be found in Appendix A.3-A.5.

**Theorem 2.** *Starting from any state $s = (s_p, s_e)$ satisfying $D(s) = d < \infty$, $\mu^*$ guarantees pursuit within $d$ steps against any evasion strategy, and $\nu^*$ avoids being captured in less than $d$ steps by any pursuit strategy.*

Based on the definition of optimal strategies in the asynchronous-move setting (see Section 2.1), Theorem 2 directly implies the following corollary:

**Corollary 1.** *For any state $s = (s_p, s_e)$ with $D(s) < \infty$, both $\mu^*$ and $\nu^*$ are optimal strategies.*

Furthermore, we use Theorem 3 to show that whether $m$ perfect-information pursuers are sufficient to capture the evader starting from state $s$ can be determined by whether $D(s) < \infty$:

**Theorem 3.** *Starting from any state $s = (s_p, s_e)$ with $D(s) = \infty$, $\nu^*$ can never be captured by any pursuit strategy.*

## 3.2 PARTIALLY OBSERVABLE SETTING

Since the DP algorithm provably generates optimal strategies when both pursuer and evader sides have full observations, it is appealing to reuse the distance table $D$ to construct a pursuit strategy under partial observability for real-world security purposes. We expect that the observation-based pursuer policy, which is extended from the DP policy under perfect information, should effectively extract history information and align with the original policy when the observation range is infinity.

We consider the following partially observable setting for the pursuers, who may serve as guards in a large area. The PEG begins because an intruder is observed, whose initial position is revealed to the pursuers. Once the game starts, the position of the evader (intruder) can no longer be detected unless it is in the observation range of at least one pursuer. For example, setting the observation range to be 2 means that the evader can be detected only when its distance to one pursuer is less than 3.

Under the partially observable setting, the observation history $o$ induces the possible positions of the evader, which we denote by a set Pos. This set is initialized as $\{s_e\}$, where $s_e$ is the initial position of the evader. As the game proceeds, it is updated based on the pursuers' observations at each timestep:

$$\text{Pos}_{\text{new}} = \begin{cases} \{s_e\} & \text{evader is observed at } s_e, \\ \text{Remove}(\text{Neighbor}(\text{Pos}_{\text{old}})) & \text{evader is not observed.} \end{cases} \tag{4}$$

where the operator Remove($\cdot$) excludes all currently observed positions (since the evader is currently unobserved) from the possible evader positions represented by Neighbor($\text{Pos}_{\text{old}}$), which corresponds to the set of one-step neighbors of the nodes in $\text{Pos}_{\text{old}}$.

Given Pos, we can express $\mu(o)$ as $\mu(s_p, \text{Pos})$ and construct a minimax policy that bounds the worst-case pursuit timesteps if we assume that the pursuers resume full observability after this step:

$$\mu(s_p, \text{Pos}) = \underset{\text{neighbor } n_p \text{ of } s_p}{\arg\min} \left\{ \max_{s_e \in \text{Pos}} \max_{\text{neighbor } n_e \text{ of } s_e} D(n_p, n_e) \right\}$$

$$= \underset{\text{neighbor } n_p \text{ of } s_p}{\arg\min} \left\{ \max_{n_e \in \text{Neighbor}(\text{Pos})} D(n_p, n_e) \right\}. \tag{5}$$

While this policy is applicable to the case of partial observability, it is based on an assumption that the observation limitation is not continual. Under continual partial observability, we find that averaging the timesteps through preserving a **belief** about the evader's position can further encourage effective pursuit, especially when the set Pos is large. The belief-averaged pursuer policy is expressed as

$$\mu(s_p, \text{belief}) = \underset{\text{neighbor } n_p \text{ of } s_p}{\arg\min} \left\{ \frac{\sum_{s_e} \text{belief}(s_e) \max_{\text{neighbor } n_e \text{ of } s_e} D(n_p, n_e)}{\sum_{s_e} \text{belief}(s_e)} \right\}, \tag{6}$$

where the belief function is initialized to be 0 except for the initial evader position and updated by

$$\text{belief}_{\text{new}}(s_e) \leftarrow \begin{cases} 0 & s_e \notin \text{Pos}, \\ \sum_{\text{neighbor } n_e \text{ of } s_e} \nu(n_e, s_e) \text{belief}_{\text{old}}(n_e) & s_e \in \text{Pos}. \end{cases} \tag{7}$$

Since the pursuer side cannot obtain the evader's policy $\nu$ when no prior knowledge is available, $\nu(n_e)$ is set to be a uniform distribution over Neighbor($n_e$) by default.

As the original DP policy $\mu^*(s)$ is provably optimal, Proposition 1 guarantees that both the position-extended policy $\mu(s_p, \text{Pos})$ and the belief-averaged policy $\mu(s_p, \text{belief})$ maintain the pursuit optimality when there is unlimited observation capability. The proof can be found in Appendix A.6.

**Proposition 1.** *When* Pos *is always a singleton, both pursuer policies (5) and (6) will be reduced to their perfect-information counterpart (1).*

Note that the time complexity of preserving Pos and belief is only $\tilde{\mathcal{O}}(|\mathcal{V}|)$ at each timestep, where $\tilde{\mathcal{O}}$ hides the additional factor of enumerating the neighbors. Since the average degree in the real-world graphs can be small (see Table 1 in Section 5), the computation is practically efficient. In Appendix B, we provide the illustrations of the belief preservation process for a more intuitive understanding.

# 4 FINDING ROBUST REAL-TIME PURSUIT STRATEGIES (R2PS) VIA ADVERSARIAL REINFORCEMENT LEARNING ACROSS GRAPHS

## 4.1 ADVERSARIAL REINFORCEMENT LEARNING

Since the DP algorithm has a lower-bound time complexity exponential in the agent number, it can be impractical to directly apply the DP policies in real time when the graph structure of the game dynamically changes. In view of this problem, we further combine our belief preservation mechanism with the idea of Equilibrium Policy Generalization (EPG) (Lu et al., 2025a) to construct a reinforcement learning method, which makes use of some preprocessed $D$ tables and the induced policies to train a generalized pursuer policy across a diverse set of graphs. We use the cross-graph RL policy for zero-shot generalization under unseen graph structures, aiming to derive worst-case robust real-time pursuit strategies (R2PS) under partial observability.

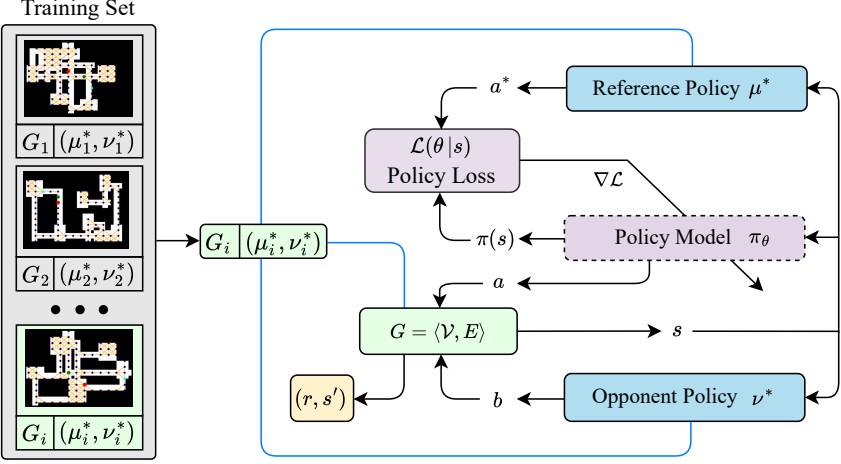

Figure 1: Cross-Graph Reinforcement Learning of Generalized Pursuer Policy

Figure 1 illustrates the cross-graph reinforcement learning pipeline, which features unexploitable evader policies as adversaries. The training set contains graphs with various topologies $G_i$ and the DP policies $(\mu_i^*, \nu_i^*)$ induced by the preprocessed $D$ tables. In each iteration, a graph $G_i$ along with the policy $(\mu_i^*, \nu_i^*)$ is sampled. Under graph $G = G_i$, we use $\mu^* = \mu_i^*$ as the reference policy to guide policy training and use $\nu^* = \nu_i^*$ as the adversarial policy. Following the principle of EPG, we train a cross-graph pursuer policy through reinforcement learning against $\nu^*$ with the guidance of $\mu^*$.

Specifically, for a transition $(s, a, b, r, s')$ in the replay buffer: $s$ is a randomly generated global state in the sampled graph; $a$ is the pursuers' joint action sampled from the current policy model $\pi_\theta$, which is ideally a graph neural network (Wu et al., 2020) with parameter $\theta$ that enables real-time inference; $b$ is the evader's action generated from the asynchronous-move opponent policy $\nu^*$ (3); the instant reward $r$ and the next state $s'$ are generated by the PEG dynamics under graph structure $G = \langle \mathcal{V}, E \rangle$.

Given state $s$, the reference policy $\mu^*$ generates a deterministic reference action $a^* = \mu^*(s)$ and serves to construct the policy loss

$$\mathcal{L}(\theta \mid s) = J_\pi(\theta \mid s) + \beta D_{\text{KL}}(\mu^*(s), \pi(s)) = J_\pi(\theta \mid s) - \beta \log \pi_\theta(s, a^*), \qquad (8)$$

where $J_\pi(\theta \mid s)$ is the original policy loss of any backbone (multi-agent) reinforcement learning algorithm (e.g., MAPPO (Yu et al., 2022)), and $\beta$ is a hyperparameter that balances policy guidance (for efficient exploration) and reinforcement learning loss (for policy optimization).

When training pursuers under partial observability, we transform the input of the policy model $\pi_\theta$ by $s \leftarrow (s_p, \text{Pos}, \text{belief})$ and use the observation-based policy $\mu(s_p, \text{Pos})$ (5) or $\mu(s_p, \text{belief})$ (6) to replace $\mu^*(s)$ (1), where Pos and belief are the preserved evader information.

Czarnecki et al. (2020) show that the strategies in real-world games have different levels of transitive strength, with Nash equilibrium being the strongest. In a fixed-graph PEG, reinforcement learning against the optimal evader policy $\nu^*$ is supposed to exclude transitively weaker pursuer policies (even if not dominated), as an equilibrium policy must act optimally against any opponent equilibrium in two-player zero-sum games. Policy guidance from $\mu^*$ further avoids accepting trivial solutions (e.g., an arbitrary policy in a Rock-Paper-Scissors game) against equilibrium. As we regard single-graph adversarial learning as policy-level division and exclusion, cross-graph training is similar to finding the joint parts of the remaining strategies under a diverse set of graph structures and abstracting them to a high-level policy. If the division criteria under different graphs are independent for the policy space due to structural distinctions, the cross-graph policy will ideally be improved at an exponential level across a diverse training corpus, leading to a generalist pursuer with "zero-shot" worst-case robustness even under partial observability.

### 4.2 Implementation and Complexity Analysis

Technically, we use soft-actor critic (SAC) (Haarnoja et al., 2018; Christodoulou, 2019) as the backbone RL algorithm and employ a decentralized architecture with a parameter-sharing graph neural network (GNN) (Cao et al., 2023; Lu et al., 2025a) to represent the graph-based policy of the homogeneous pursuers. The SAC algorithm features a self-adaptive entropy regularization that balances exploration and exploitation, with double Q-learning (Hasselt, 2010) employed to avoid overestimation. The GNN architecture combines multi-head self-attention (Vaswani et al., 2017) with adjacent-matrix masks to encode graph-based states. The state embedding is then sent into a decoder followed by a pointer network (Vinyals et al., 2015) for graph-based policy output.

The implementation details and hyperparameter setting are reserved in Appendix C to save space[1]. According to the corresponding analysis, the overall time complexity of computing the graph-based state feature is $\mathcal{O}(n^2m)$, where $n = |\mathcal{V}|$ is the number of vertices in the graph, and $m$ is the number of pursuers. Since the complexity of GNN queries is also $\mathcal{O}(n^2m)$, and the complexity of preserving Pos and belief is $\tilde{\mathcal{O}}(n)$, the overall inference time complexity of the RL pursuer policy at each timestep is only $\mathcal{O}(n^2m) + \mathcal{O}(n^2m) + \tilde{\mathcal{O}}(n) = \mathcal{O}(n^2m)$. In comparison, the time complexity of recomputing DP policies is $\tilde{\mathcal{O}}(n^{m+1})$ under dynamically changing graph structures (see Lu et al. (2025a)), as Algorithm 1 needs to be repeatedly executed. Here we briefly show the inference time gap arising from this complexity distinction. When $n = 1000$ and $m = 2$, it takes over 2 minutes to run Algorithm 1 at each timestep using an Intel Core i9-13900HX CPU. The inference time of our GNN-represented RL policy, however, is less than 1 second under the same condition. Under GPU accelerations, the time can be further reduced to below 0.01 seconds (see Section 5.3).

## 5 Evaluations

Here we provide our experimental evaluations of single-graph DP pursuers and cross-graph RL pursuers under partial observability. We assume that there are two pursuers ($m = 2$) against the single evader. This is a reasonable setting in view of the graph-theoretic result that 3 pursuers with full observations can always capture the evader in any planar graph (Fromme & Aigner, 1984). The initial position is randomly generated under the restriction that the distance between the evader and the pursuers is larger than the observation range of 2. Besides, no observation sensors except for the

---

[1] Code can be found at https://github.com/Cahemgco/EPG_code.

pursuers themselves are allowed. The test graphs include Grid Map (a $10 \times 10$ grid), Scotland-Yard Map (from the board game Scotland-Yard), Downtown Map (a real-world location from Google Maps), and 7 famous real-world spots (from Times Square to Sydney Opera House). The graph details are shown in Appendix D.1, and the statistics of these graphs are shown in Table 1 (left).

Table 1: Graph Data (Total Node Number, Average Degree, Diameter) and Success Rate Comparison

|  | Node | Degree | Diameter | Shortest Path | $DP_{Pos}$ | $DP_{belief}$ |
|---|---|---|---|---|---|---|
| Grid Map | 100 | 3.60 | 18 | 0.00 | 0.59 | **0.78** |
| Scotland-Yard Map | 200 | 3.91 | 19 | 0.00 | 0.44 | **0.63** |
| Downtown Map | 206 | 2.98 | 19 | 0.02 | 0.73 | **0.90** |
| Times Square | 171 | 2.58 | 22 | 0.01 | 0.41 | **0.69** |
| Hollywood Walk of Fame | 201 | 2.42 | 31 | 0.01 | 0.25 | **0.48** |
| Sagrada Familia | 231 | 2.60 | 25 | 0.00 | 0.24 | **0.36** |
| The Bund | 200 | 2.53 | 29 | 0.03 | 0.30 | **0.57** |
| Eiffel Tower | 202 | 2.34 | 38 | 0.29 | 0.69 | **0.94** |
| Big Ben | 192 | 2.48 | 34 | 0.08 | 0.54 | **0.74** |
| Sydney Opera House | 183 | 2.33 | 37 | 0.05 | 0.47 | **0.87** |

## 5.1 EVALUATIONS OF EXTENDED DP PURSUERS

We first evaluate the strength of the extended DP pursuers under partial observability (Section 3.2). We denote by $DP_{Pos}$ the position-extended pursuer (5) and by $DP_{belief}$ the belief-averaged pursuer (6). The pursuers succeed ($f(s) = 1$) when at least one of them is adjacent to the evader on the graph within $128$ timesteps, and the success rates are averaged over $500$ tests. To simulate the difficult case for security purposes, the evader is set to be the provably optimal DP evader (3) with global observations and asynchronous moves. For an intuitive comparison, we also include the result of directly following the shortest path to the evader under full observability.

As is shown in Table 1 (right), the shortest-path strategy can hardly capture the optimal DP evader. In comparison, though under a limited observation range of 2, the extended DP pursuers demonstrate significantly higher success rates. Besides, $DP_{belief}$ consistently outperforms $DP_{Pos}$. This result verifies that the direct minimax policy (5) can be improved through belief averaging. Actually, since equation (5) treats all possible positions as equal, the result of the inner max can be very large when the size of Pos is large, leading to pessimistic pursuit behaviors like staying at certain "rest points."

We further take a look at how observation capabilities could affect success rates. We increase the observation range and evaluate the performance of $DP_{belief}$ (6). As is shown in Table 6 (Appendix D.2), the success rates monotonically increase with the observation range and reach $100\%$ when the range exceeds 5. While $D(\cdot)$ is an accurate estimator of the worst-case pursuit distance in Markov PEGs, it becomes an optimistic one under partial observability. Nevertheless, the experimental results show that combining this optimistic estimator with belief information can maintain the strength of the DP-based pursuit strategies, even under very limited observation capabilities.

## 5.2 EVALUATIONS OF GENERALIZED RL PURSUERS

Now, we implement and evaluate our cross-graph reinforcement learning method aimed at R2PS (Section 4). We discretize the maps from the Dungeon environment (Chen et al., 2019) to construct a synthetic training set containing 150 graphs and further include 150 random urban locations from Google Maps to create a large training set with a total of 300 graphs, where the maximum node number is no more than 500. We apply the R2PS learning scheme to the synthetic training set and the large training set. Appendix C.4 provides the learning curves of the pursuer policies under partial observability. As is shown in Figure 4, using the extended DP pursuers as guidance ($\beta = 0.1$) helps to improve the training efficiency over pure reinforcement learning ($\beta = 0$) under either training set.

Policy-Space Response Oracles (PSRO) (Lanctot et al., 2017) is a general reinforcement learning method extended from the game-theoretic approach of double oracle (DO) (McMahan et al., 2003) for equilibrium finding. Here we compare the zero-shot performance of our generalized pursuer policy with a PSRO policy that is directly trained on the 10 test graphs using 10 iterations (10000

Table 2: Success Rate Comparison across Different Graphs and Strategies

| Evader Policy | Stay | | $DP_{sync}$ | | $DP_{async}$ | | $BR_{async}$ |
|---|---|---|---|---|---|---|---|
| Pursuer Policy | Ours | PSRO | Ours | PSRO | Ours | PSRO | Ours |
| Grid Map | **1.00** | **1.00** | **1.00** | 0.94 | **1.00** | 0.88 | 1.00 |
| Scotland-Yard Map | **1.00** | **1.00** | **1.00** | 0.47 | **0.76** | 0.00 | 0.73 |
| Downtown Map | **1.00** | 0.99 | **1.00** | 0.88 | **0.99** | 0.03 | 0.92 |
| Times Square | **1.00** | 0.93 | **1.00** | 0.16 | **0.95** | 0.04 | 0.27 |
| Hollywood Walk of Fame | **1.00** | 0.95 | **0.90** | 0.00 | **0.38** | 0.00 | 0.10 |
| Sagrada Familia | 0.99 | 0.93 | **0.96** | 0.07 | **0.20** | 0.00 | 0.20 |
| The Bund | **1.00** | 0.95 | **0.92** | 0.31 | **0.25** | 0.04 | 0.23 |
| Eiffel Tower | **1.00** | 0.99 | **1.00** | 0.97 | **1.00** | 0.52 | 0.55 |
| Big Ben | **1.00** | 0.99 | **1.00** | 0.29 | **0.82** | 0.24 | 0.65 |
| Sydney Opera House | **1.00** | 0.98 | **1.00** | 0.07 | **0.95** | 0.11 | 0.31 |

episodes per iteration). Our RL policy aimed at R2PS, however, is pretrained under the synthetic training set with 150 graphs for 30000 episodes ($\beta = 0.1$) and then trained under the 150 random urban graphs for 70000 episodes. Since our training process never comes across the test graphs, our RL policy has to zero-shot generalize to these unseen graph structures during evaluations.

As is shown in Table 2, our pursuer policy consistently outperforms the PSRO pursuer policy in the real-world graphs against a variety of opponents, where:

- Stay corresponds to an evader that stays at the initial position. Since the initial distance between the pursuers and the evader is larger than the observation range, and the pursuers have no prior knowledge about the evader's policy, staying still is a reasonable strategy and leads to the occasional failure of these RL pursuers.

- $DP_{sync}$ corresponds to the DP evader policy (2) under synchronous moves, and $DP_{async}$ corresponds to the strictly optimal policy (3) under asynchronous moves. It is clear that the asynchronous-move evaders are much stronger than the synchronous-move ones due to the advantage of forecasting the pursuers' decisions. Against $DP_{async}$, the PSRO pursuers struggle under most of the test graphs in comparison with ours.

- $BR_{async}$ corresponds to the best-responding asynchronous-move evader directly trained against our RL pursuers in the test graphs for 30000 episodes (converged). Even under this worst case, the success rates of our generalized pursuers are over $50\%$ in half of the graphs.

Since our worst-case zero-shot performance is clearly better than the PSRO policy directly trained on the test graphs, we can say that our real-time strategies are worst-case robust even under varying graph structures, which implies that our approach achieves R2PS under partial observability.

## 5.3 Scalability Tests and Ablation Studies

Table 3: RL Success Rate (against $DP_{async}$) and Comparison of Inference Time in Large Graphs

| | Node Number | Success Rate | RL Time (s) | DP Time (s) |
|---|---|---|---|---|
| Times Square | 1805 | 0.56 | 0.009837 | 101 |
| Hollywood Walk of Fame | 1251 | 0.46 | 0.007917 | 33 |
| Sagrada Familia | 2065 | 0.33 | 0.009895 | 139 |
| The Bund | 1723 | 0.46 | 0.008117 | 83 |
| Eiffel Tower | 1825 | 0.41 | 0.009616 | 96 |
| Big Ben | 1681 | 0.49 | 0.007752 | 79 |
| Sydney Opera House | 744 | 0.76 | 0.007648 | 6 |

Now we further verify the real-time pursuit capability under the graphs with higher complexity. We create another set of test graphs based on the seven famous locations in Table 1 (from Times Square to Sydney Opera House). Compared to the original graphs, the new graphs double both

the map range and the discretization accuracy, leading to significantly larger node numbers. The success rates of our RL pursuer policy against the optimal evader $DP_{async}$ and the inference time comparisons under an NVIDIA GeForce RTX 2080 Ti GPU are shown in Table 3. Clearly, our RL policy requires significantly smaller inference time in comparison with DP and maintains desirable overall performance under large graphs in comparison with the results in Table 2. Figure 6 (Appendix D.2) provides the scaling plots of our GNN-based RL policy inference and DP computation time.

We are also curious about whether our RL policy trained under the limited observation range of 2 can demonstrate better performance when the observation range is larger during inference time. As is shown in Table 7 (Appendix D.2), the success rates of our RL pursuers monotonically increase with the observation range. This additional result implies that our RL policy trained with the minimum observability can be directly applied to the cases with better sensing capabilities.

Finally, we examine how the belief updates affect pursuit performance. As we have mentioned, our belief preservation (7) always employs a uniform evader policy $\nu$ since we could not access prior information about the true opponent. However, if we manage to obtain such information in reality, we can instantly improve the pursuit performance by replacing $\nu$ with the actual evader policy. As is shown in Table 4, utilizing known opponent information improves success rates against the best-responding evader $BR_{async}$. On the other hand, if we reduce the belief update frequency from every single step (original) to every 2 or 3 steps, then the pursuit success rates will instantly decline. This result further demonstrates the benefits of our belief update mechanism.

Table 4: RL Success Rate (against $BR_{async}$) Comparison under Different Belief Update Conditions

| Belief Update Condition | Known Opponent | Original | Every 2 Steps | Every 3 Steps |
|---|---|---|---|---|
| Grid Map | 1.00 | 1.00 | 0.60 | 0.42 |
| Scotland-Yard Map | 0.99 | 0.73 | 0.34 | 0.28 |
| Downtown Map | 1.00 | 0.92 | 0.61 | 0.39 |
| Times Square | 0.42 | 0.27 | 0.18 | 0.17 |
| Hollywood Walk of Fame | 0.13 | 0.10 | 0.04 | 0.03 |
| Sagrada Familia | 0.28 | 0.20 | 0.12 | 0.05 |
| The Bund | 0.54 | 0.23 | 0.13 | 0.12 |
| Eiffel Tower | 0.81 | 0.55 | 0.32 | 0.29 |
| Big Ben | 0.82 | 0.65 | 0.40 | 0.25 |
| Sydney Opera House | 0.54 | 0.31 | 0.22 | 0.15 |

## 6 CONCLUSION

This paper presents a general approach to worst-case robust real-time pursuit strategies under partial observability and varying graph structures. We first theoretically examine a dynamic programming (DP) algorithm and prove that it can unify the solutions to Markov PEGs with either synchronous moves or asynchronous moves. Then, we propose a belief preservation mechanism to efficiently abstract evader information from the observation histories of the pursuers and thus construct the observation-based pursuer policies. Finally, we embed the belief preservation mechanism into the framework of EPG (Lu et al., 2025a) to find robust real-time pursuit strategies, fulfilling cross-graph reinforcement learning against the asynchronous-move DP evader under partial observability. Experiments show that our observation-based DP pursuers can be used as guidance to facilitate efficient policy exploration during RL training. Under unseen real-world graph structures, our cross-graph policy manages to generate real-time pursuit strategies with worst-case robustness, consistently outperforming the PSRO policy directly trained under the test graphs. Comparative results also reveal that the pursuers can benefit from belief updates, while the evader benefits from asynchronous moves.

In this work, the belief preservation mechanism provides an affordable way to handle partial observability in real time. We show that this mechanism can be effectively combined with the existing PEG methods like DP and EPG. After adversarial reinforcement learning across graphs, a generalized pursuer policy under belief preservation is eventually derived, leading to the first worst-case robust real-time pursuit strategies under partial observability. Hopefully, the current research on PEGs could encourage subsequent works on the broader research topics concerning real-world security.

ACKNOWLEDGMENTS

This work was supported in part by the National Natural Science Foundation of China under Grants 62293541 and 62136008 and in part by the Beijing Nova Program under Grant 20240484514. We also thank all the reviewers for their helpful suggestions during the ICLR review process.

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

## A OMITTED PROOFS

### A.1 PROOF OF THEOREM 1

*Proof.* For no-exit PEGs, the Nash value satisfies the following Bellman minimax equation:

$$
V^*(s) = \begin{cases} \max_{\mu(s) \in \Delta(\mathcal{A})} \min_{b \in \mathcal{B}} \sum_{a \in \mathcal{A}} \mu(s,a) \left( r(s,a,b) + \gamma \sum_{s' \in S} \mathcal{P}(s,a,b,s') V^*(s') \right) & f(s) = 0, \\ 1 & f(s) = 1. \end{cases}
$$

Since the transition is deterministic and a non-zero reward is received only when a termination state is reached, we can simplify the Bellman equation as follows:

$$
V^*(s) = \begin{cases} \max_{\mu(s) \in \Delta(\mathcal{A})} \min_{b \in \mathcal{B}} \sum_{a \in \mathcal{A}} \mu(s,a) \gamma V^*(s' = \mathcal{P}(s,a,b)) & f(s) = 0, \\ 1 & f(s) = 1. \end{cases}
$$

The equilibrium policy for the max-player satisfies:

$$
\mu^*(s) \in \underset{\mu(s) \in \Delta(\mathcal{A})}{\arg\max} \left\{ \min_{b \in \mathcal{B}} \sum_{a \in \mathcal{A}} \mu(s,a) V^*(s' = \mathcal{P}(s,a,b)) \right\}.
$$

When there is a pure-strategy Nash equilibrium in the game, the $\arg\max$ has a pure-strategy solution, and the Bellman equation can be further simplified:

$$
V^*(s) = \gamma \max_{a \in \mathcal{A}} \min_{b \in \mathcal{B}} V^*(s' = \mathcal{P}(s,a,b)). \tag{9}
$$

Note that the Nash value has the form of $V^*(s) = \gamma^d (d \in \mathbb{N})$. Therefore, we consider using mathematical induction. We assume that $V^*(s) = \gamma^{D(s)}$ holds for all states $s$ that satisfies either $V^*(s) = \gamma^d$ or $D(s) = d$ when $d < k$ ($\gamma^d > \gamma^k$). We want to prove that $V^*(s) = \gamma^{D(s)}$ holds for all states $s$ that satisfies either $V^*(s) = \gamma^k$ or $D(s) = k$. Clearly, our initialization guarantees that the proposition holds for $k = 0$. Our update condition $D(n_p, n_e) = \infty$ guarantees that every state $s \in S$ is pushed into and popped from $\mathcal{Q}$ at most once. Note that the following proof reverses the notations of $s$ and $s'$ in (9) to better align with $s = (s_p, s_e)$ in Algorithm 1.

Now, we prove the first half of the proposition. For an arbitrary state $s' = (n_p, n_e)$ that satisfies $V^*(s') = \gamma^k$, the simplified Bellman equation (9) guarantees that there exists $a = s_p \in \mathcal{A}(n_p)$ and $b = s_e \in \mathcal{B}(n_e)$ such that $V^*(s') = \gamma V^*(s = \mathcal{P}(s', a, b))$. Therefore, there exists $s = (s_p, s_e)$ such that $V^*(s) = \gamma^{k-1}$. According to the first half of the induction hypothesis, we have that $D(s) = k - 1 < \infty$, which implies that the algorithm once pushed $s'$ into $\mathcal{Q}$. Besides, the Bellman equation guarantees that $\forall b' \in \mathcal{B}(n_e), V^*(\mathcal{P}(s', a, b')) \geq V^*(\mathcal{P}(s', a, b)) = V^*(s) = \gamma^{k-1} > \gamma^k$. By induction hypothesis, $D(s_p, n'_e) \leq D(s_p, s_e)$ holds for any neighbor $n'_e$ of $n_e$. Therefore, the algorithm must enumerate $n_e$ when popping $s = (s_p, s_e)$. If we have $D(n_p, n_e) = \infty$ at the moment, then $n_p$ will be enumerated in the inner loop, and we will have $D(n_p, n_e) = D(s_p, s_e) + 1 = k$. Now we complete the proof by showing that $D(n_p, n_e) < \infty$ implies $D(n_p, n_e) = k$. Actually, if $k < D(n_p, n_e) < \infty$, then $D(s')$ must be computed by adding 1 to some $D(s'') \geq k$. Since $s''$ must be popped from $\mathcal{Q}$ no later than $s$, it is contradictory to the fact that $D(s'') > D(s) = k - 1$. If $D(n_p, n_e) < k$, then the second half of the induction hypothesis implies that $V^*(s') = \gamma^{D(n_p, n_e)}$, which is contradictory to the fact that $V^*(s') = \gamma^k$.

Then, we prove the second half of the proposition. For an arbitrary state $s' = (n_p, n_e)$ that satisfies $D(s') = k$, $D(s')$ must be computed by adding 1 to some $D(s) = k - 1$, where $s = (s_p, s_e)$. According to the second half of the induction hypothesis, we have $V^*(s) = \gamma^{k-1}$. The algorithm guarantees that $D(s_p, n'_e) \leq D(s_p, s_e) = k - 1$ holds for any neighbor $n'_e$ of $n_e$. By induction hypothesis, it holds that $\forall b' \in \mathcal{B}(n_e), V^*(\mathcal{P}(s', a, b')) \geq V^*(\mathcal{P}(s', a, b))$ when $a = s_p \in \mathcal{A}(n_p)$ and $b = s_e \in \mathcal{B}(n_e)$. Therefore, $\min_{b \in \mathcal{B}(n_e)} V^*(\mathcal{P}(s', a, b)) = \gamma^{k-1}$ when $a = s_p \in \mathcal{A}(n_p)$. If there exists $a^\dagger = s_p^\dagger \in \mathcal{A}(n_p)$ such that $\min_{b \in \mathcal{B}(n_e)} V^*(\mathcal{P}(s', a', b)) > \gamma^{k-1}$,

then we let $b^\dagger = \arg\min\limits_{b \in \mathcal{B}(n_e)} V^*(\mathcal{P}(s', a^\dagger, b)) > \gamma^{k-1}$ and let $s^\dagger = (s_p^\dagger, s_e^\dagger = b^\dagger)$. According to the first half of the induction hypothesis, $D(s_p^\dagger, n_e') \leq D(s_p^\dagger, s_e^\dagger) < k - 1$ holds for any neighbor $n_e'$ of $n_e$. Since $D(s_p^\dagger, s_e^\dagger) < D(s_p, s_e)$, $s^\dagger$ must be popped from $\mathcal{Q}$ earlier than $s$. This is contradictory to the fact that $D(s') = \infty$ when $s$ is popped: If it holds, then $s' = (n_p, n_e)$ must have been enumerated when $s^\dagger$ is popped. Therefore, $V^*(s') = \gamma \max\limits_{a \in \mathcal{A}} \min\limits_{b \in \mathcal{B}} V^*(\mathcal{P}(s', a, b)) = \gamma^k$.

For now, we have proved that $V^*(s) = \gamma^{D(s)}$. Therefore:

$$\mu^*(s_p, s_e) = \underset{\text{neighbor } n_p \text{ of } s_p}{\arg\min} \left\{ \max_{\text{neighbor } n_e \text{ of } s_e} D(n_p, n_e) \right\} \Rightarrow \mu^*(s) = \underset{a \in \mathcal{A}}{\arg\max} \min_{b \in \mathcal{B}} V^*(\mathcal{P}(s, a, b)),$$

$$\nu^*(s_p, s_e) = \underset{\text{neighbor } n_e \text{ of } s_e}{\arg\max} \left\{ \min_{\text{neighbor } n_p \text{ of } s_p} D(n_p, n_e) \right\} \Rightarrow \nu^*(s) = \underset{b \in \mathcal{B}}{\arg\min} \max_{a \in \mathcal{A}} V^*(\mathcal{P}(s, a, b)).$$

As there exists a pure-strategy Nash equilibrium, it is directly guaranteed that $(\mu^*, \nu^*)$ is a Nash equilibrium. $\qquad\square$

## A.2 Proof of Lemma 1

*Proof.* We consider the cases of $D(n_p, n_e) = \infty$ and $0 < D(n_p, n_e) < \infty$, separately.

The first case is $D(n_p, n_e) = \infty$, which implies that $(n_p, n_e)$ is never enqueued:

Suppose that $\min\limits_{\text{neighbor } s_p \text{ of } n_p} \left\{ \max\limits_{\text{neighbor } s_e \text{ of } n_e} D(s_p, s_e) \right\} < \infty$. Then, we let

$$s_p = \underset{\text{neighbor } s_p \text{ of } n_p}{\arg\min} \left\{ \max\limits_{\text{neighbor } s_e \text{ of } n_e} D(s_p, s_e) \right\}, s_e = \underset{\text{neighbor } s_e \text{ of } n_e}{\arg\max} D(s_p, s_e).$$

Since $D(s_p, s_e) = \min\limits_{\text{neighbor } s_p \text{ of } n_p} \left\{ \max\limits_{\text{neighbor } s_e \text{ of } n_e} D(s_p, s_e) \right\} < \infty$, $(s_p, s_e)$ is once enqueued.

Since $s_e = \underset{\text{neighbor } s_e \text{ of } n_e}{\arg\max} D(s_p, s_e)$, we have that $\nexists n_e' \in \mathcal{V}, (n_e, n_e') \in E, D(s_p, n_e') > D(s_p, s_e)$.
Since $D(n_p, n_e) = \infty$, state $(n_p, n_e)$ will be enumerated when $(s_p, s_e)$ is dequeued. Then, $(n_p, n_e)$ is enqueued, which leads to a contradiction.

Therefore, we have $\min\limits_{\text{neighbor } s_p \text{ of } n_p} \left\{ \max\limits_{\text{neighbor } s_e \text{ of } n_e} D(s_p, s_e) \right\} = \infty$, which means the equation holds in the first case.

The second case is $0 < D(n_p, n_e) < \infty$, which implies that $(n_p, n_e)$ is once enumerated when a state $(s_p, s_e) \in \text{Neighbor}(n_p, n_e)$ is dequeued. According to the enumeration rule, we have $\nexists n_e' \in \mathcal{V}, (n_e, n_e') \in E, D(s_p, n_e') > D(s_p, s_e)$, which implies $s_e = \underset{\text{neighbor } s_e \text{ of } n_e}{\arg\max} D(s_p, s_e)$. Since $D(n_p, n_e) = D(s_p, s_e) + 1$, we have:

$$D(n_p, n_e) \geq \min\limits_{\text{neighbor } s_p \text{ of } n_p} \left\{ \max\limits_{\text{neighbor } s_e \text{ of } n_e} D(s_p, s_e) \right\} + 1.$$

Now redefine $s_p = \underset{\text{neighbor } s_p \text{ of } n_p}{\arg\min} \left\{ \max\limits_{\text{neighbor } s_e \text{ of } n_e} D(s_p, s_e) \right\}, s_e = \underset{\text{neighbor } s_e \text{ of } n_e}{\arg\max} D(s_p, s_e).$

Then, we have $D(s_p, s_e) = \min\limits_{\text{neighbor } s_p \text{ of } n_p} \left\{ \max\limits_{\text{neighbor } s_e \text{ of } n_e} D(s_p, s_e) \right\}$ and $\nexists n_e' \in \mathcal{V}, (n_e, n_e') \in E, D(s_p, n_e') > D(s_p, s_e)$. Since the $D$ values in $\mathcal{Q}$ do not decrease, we have $D(n_p, n_e) \leq D(s_p, s_e) + 1 = \min\limits_{\text{neighbor } s_p \text{ of } n_p} \left\{ \max\limits_{\text{neighbor } s_e \text{ of } n_e} D(s_p, s_e) \right\} + 1.$

Therefore, $D(n_p, n_e) = \min\limits_{\text{neighbor } s_p \text{ of } n_p} \left\{ \max\limits_{\text{neighbor } s_e \text{ of } n_e} D(s_p, s_e) \right\} + 1$ holds in the second case.

To conclude, the equation always holds when $D(n_p, n_e) > 0$. $\qquad\square$

## A.3 Proof of Theorem 2

*Proof.* First, we prove that for any state $s = (s_p, s_e)$ satisfying $D(s) = d < \infty$, $\mu^*$ guarantees pursuit within $d$ steps against any evasion strategy:

Clearly, the proposition holds true when $d = 0$. Assume that the proposition holds for all $d < k$. When $d = k$, we let $n_p = \underset{\text{neighbor } n_p \text{ of } s_p}{\arg \min} \left\{ \underset{\text{neighbor } n_e \text{ of } s_e}{\max} D(n_p, n_e) \right\}, n_e = \underset{\text{neighbor } n_e \text{ of } s_e}{\arg \max} D(n_p, n_e)$.
By Lemma 1, we have $D(n_p, n'_e) \leq D(n_p, n_e) = D(s_p, s_e) - 1 \leq k - 1 < k, \forall n'_e \in \text{Neighbor}(s_e)$. By induction hypothesis, $\mu^*$ guarantees pursuit within $k - 1$ steps for the states $(n_p, n'_e)$ that satisfies $n'_e \in \text{Neighbor}(s_e)$. Therefore, $\mu^*(s) = n_p$ guarantees pursuit within $k$ steps. By induction, the proposition holds true for all $d < \infty$.

Second, we prove that for any state $s = (s_p, s_e)$ satisfying $D(s) = d < \infty$, $\nu^*$ avoids being captured in less than $d$ steps by any pursuit strategy:

Suppose there exists a pursuit movement sequence $\{n_p^0, n_p^1, \cdots, n_p^{T-1}\}$ that captures the evader within $T < d$ steps under policy $\nu^*$. Then, we denote by $\{s^0, s^1, s^2, \cdots, s^T\}$ the corresponding state sequence, where $s_0 = s$ and $D(s^T) = 0$. By Lemma 1, $D(s_p, s_e) = \underset{\text{neighbor } n_p \text{ of } s_p}{\min} \{\nu^*(s_p, s_e, n_p)\} + 1$, which implies that $D(s_p, s_e) \geq D(n_p, \nu^*(s_p, s_e, n_p)) + 1, \forall n_p$. Therefore, $D(s^t) = D(s_p^t, s_e^t) \geq D(n_p^t, \nu^*(s_p^t, s_e^t, n_p^t)) + 1 = D(s_p^{t+1}, s_e^{t+1}) + 1 = D(s^{t+1}) + 1$. This leads to a contradiction: $D(s) = D(s^0) \geq D(s^1) + 1 \geq D(s^2) + 2 \geq D(s^T) + T = T < d = D(s)$. Therefore, $\nu^*$ always avoids being captured in less than $d$ steps when $D(s) = d < \infty$. □

## A.4 Proof of Corollary 1

*Proof.* By Theorem 2:

For $s = (s_p, s_e)$ with $D(s) = d < \infty$, since $\mu^*$ guarantees pursuit within $d$ steps against any evasion strategy, no evader can guarantee evasion for $d$ steps against a worst-case pursuit strategy. Since $\nu^*$ avoids being captured in less than $d$ steps by any pursuit strategy, $\nu^*$ is the optimal evasion strategy.

For $s = (s_p, s_e)$ with $D(s) = d < \infty$, since $\nu^*$ avoids being captured in less than $d$ steps by any pursuit strategy, no pursuer can guarantee pursuit in less than $d$ steps against a worst-case evasion strategy. Since $\mu^*$ guarantees pursuit within $d$ steps against any evasion strategy, $\mu^*$ is the optimal pursuit strategy. □

## A.5 Proof of Theorem 3

*Proof.* Suppose that there exists a pursuit strategy that captures $\nu^*$ within $T$ steps.

In this case, we denote by $\{s^0, s^1, s^2, \cdots, s^T\}$ the state sequence of a successful pursuit, where $s_0 = s$ and $D(s^T) = 0$. By Lemma 1, $D(s_p, s_e) = \underset{\text{neighbor } n_p \text{ of } s_p}{\min} \{D(n_p, \nu^*(n_p))\} + 1$, which implies $D(n_p, \nu^*(n_p)) = \infty, \forall n_p \in \text{Neighbor}(s_p)$. Then, we have $D(s^0) = \infty \Rightarrow D(s^1) = \infty \Rightarrow \cdots \Rightarrow D(s^T) = \infty$, which leads to a contradiction.

Therefore, for $s = (s_p, s_e)$ with $D(s) = \infty$, $\nu^*$ can never be captured by any pursuit strategy. □

## A.6 PROOF OF PROPOSITION 1

*Proof.* When $\mathrm{Pos}$ is a singleton, we have $\mathrm{Pos} = \{s_e\}$. Therefore,

$$
\mu(s_p, \mathrm{Pos}) = \underset{\text{neighbor } n_p \text{ of } s_p}{\arg\min} \left\{ \max_{n_e \in \mathrm{Neighbor(Pos)}} D(n_p, n_e) \right\}
$$

$$
= \underset{\text{neighbor } n_p \text{ of } s_p}{\arg\min} \left\{ \max_{\text{neighbor } n_e \text{ of } s_e} D(n_p, n_e) \right\} = \mu^*(s_p, s_e).
$$

Besides, we have

$$
\mathrm{belief(s)} = \begin{cases} 0 & s \notin \mathrm{Pos} \\ \displaystyle\sum_{\text{neighbor } v \text{ of } s} \nu(v, s) \mathrm{belief_{old}(v)} & s \in \mathrm{Pos} \end{cases}
$$

$$
= \mathbb{I}\left[s = s_e\right] \mathrm{belief_{new}(s_e)}.
$$

Therefore, it holds that

$$
\mu(s_p, \mathrm{belief}) = \underset{\text{neighbor } n_p \text{ of } s_p}{\arg\min} \left\{ \frac{\mathrm{belief_{new}(s_e)} \displaystyle\max_{\text{neighbor } n_e \text{ of } s_e} D(n_p, n_e)}{\mathrm{belief_{new}(s_e)}} \right\}
$$

$$
= \underset{\text{neighbor } n_p \text{ of } s_p}{\arg\min} \left\{ \max_{\text{neighbor } n_e \text{ of } s_e} D(n_p, n_e) \right\} = \mu^*(s_p, s_e).
$$

$\square$

# B    ILLUSTRATION OF BELIEF PRESERVATION

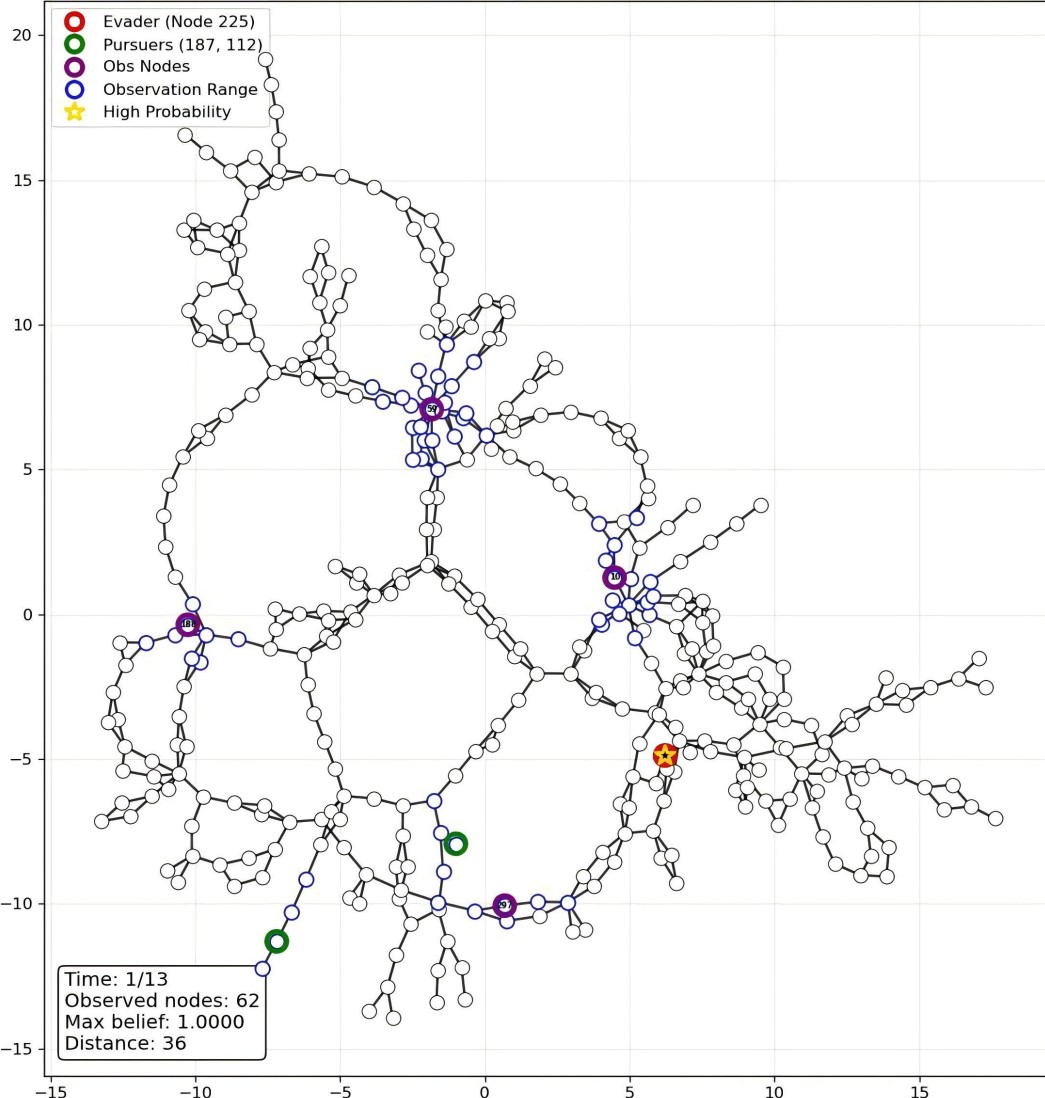

Figure 2: Pursuit Initialization under Limited Observation Range (Nodes with Blue Outlines)

Figure 2 illustrates the initial state of one pursuit episode. Two green pursuers with an observation range of 2 are going to capture the red evader that stays still within a total of 13 steps. The purple nodes are some auxiliary sensors that provide additional information for the pursuers (also with an observation range of 2). Therefore, all of the nodes with dark blue outlines can be observed by the pursuers, while the other nodes cannot. At the first step, only the red node has non-zero belief and is marked as high probability (represented by the yellow star).

Figure 3 illustrates the pursuit process under belief preservation, where the black or shadowed area around the evader corresponds to belief and the darkness of the nodes indicates the current belief distribution. Following (7), the belief is spread as the game proceeds (see timesteps $2, 4, 6, 8, 10$) and used to generate the pursuit strategy (6) under partial observability. At timestep 12, the evader is eventually observed. Since Pos becomes a singleton by (4), the shadowed area disappears, and the observed evader node is marked as high probability again.

Note that when the observed area covers all nodes, the game will be reduced to its perfect-information counterpart, where the DP pursuer policy is provably optimal.

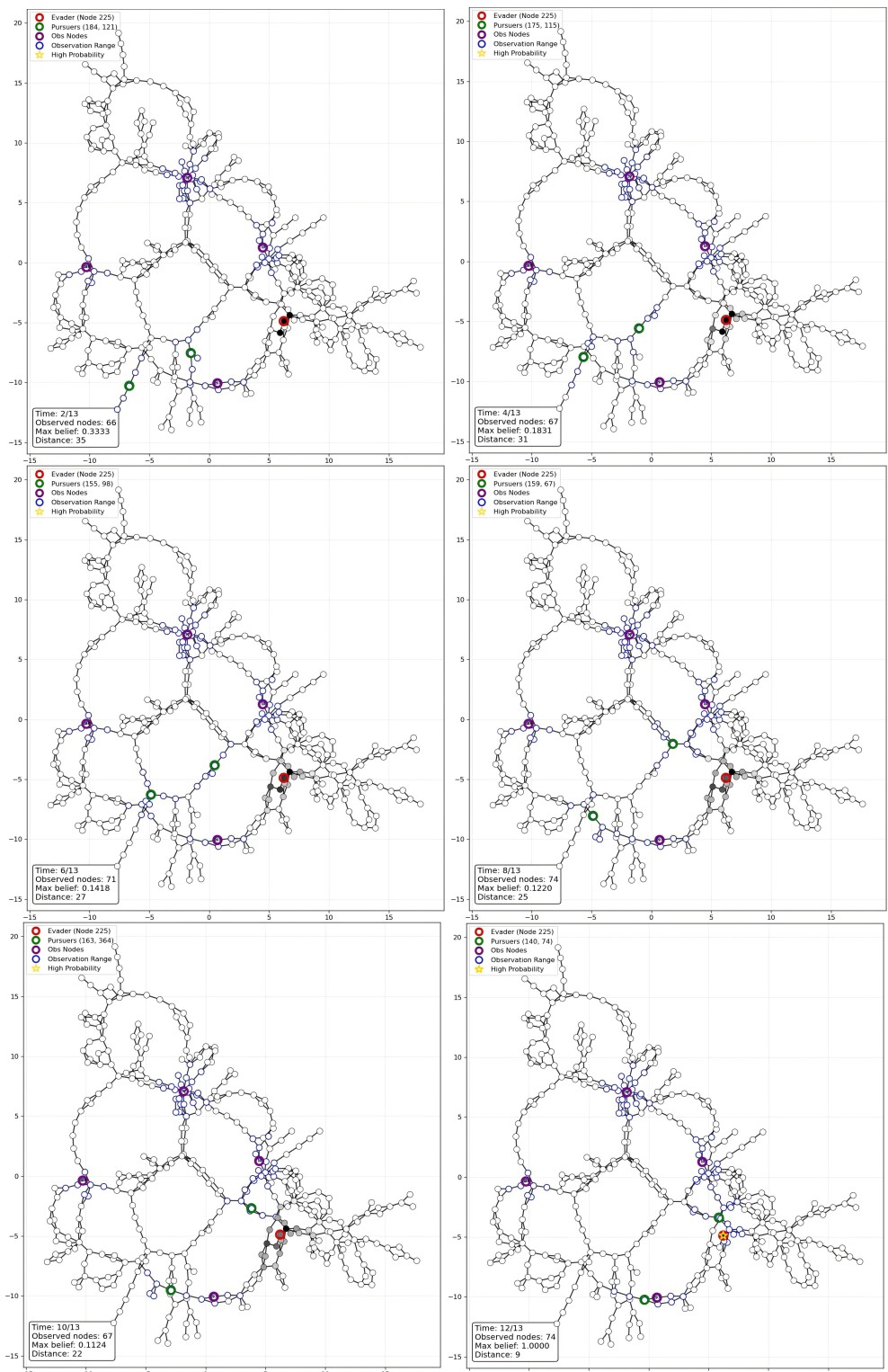

Figure 3: Pursuit Illustration under Belief Preservation (Shadowed Area around Evader)

## C  IMPLEMENTATION DETAILS

### C.1  SOFT ACTOR-CRITIC (SAC)

To fulfill R2PS training, we use discrete-action soft-actor critic (SAC) (Christodoulou, 2019) as the backbone RL algorithm, where:

The value function is defined as

$$V(s) = \mathbb{E}_{a \sim \pi(s)} \left[ Q(s, a) - \alpha \log \pi(s, a) \right].$$

The loss of the value network $Q_\phi$ is computed as

$$J_Q(\phi) = \mathbb{E}_{s,a} \left[ \frac{1}{2} (Q_\phi(s, a) - (r + \gamma \mathbb{E}_{s'} \left[ V(s') \right]))^2 \right].$$

The loss of the policy network $\pi_\theta$ is computed as

$$J_\pi(\theta) = \mathbb{E}_{s,a \sim \pi_\theta(s)} \left[ \alpha \log \pi_\theta(s, a) - Q(s, a) \right].$$

The temperature $\alpha$ under target entropy $\overline{H}$ is adaptively updated under loss

$$J(\alpha) = \mathbb{E}_{s,a} \left[ -\alpha \left( \log \pi(s, a) + \overline{H} \right) \right].$$

### C.2  GRAPH NEURAL NETWORK (GNN)

We employ a sequence model with a parameter-sharing graph neural network (GNN) architecture (Lu et al., 2025a) to represent the graph-based policy of the homogeneous pursuers:

Under the principle of sequential decision-making, a joint policy can be decomposed as

$$\pi(a_1, a_2, \cdots, a_m | s) = \prod_{l=1}^{m} \pi(a_l | s, a_1, \cdots, a_{l-1}),$$

where $(s, a_1, \cdots, a_{l-1})$ indicates the global state after the first $l - 1$ pursuers take actions $(a_i)_{i \in [l-1]}$.

For a team of pursuers with $m$ agents, the sequence model queries the policy network $m$ times under a fixed adjacent matrix $M \in \{0, 1\}^{n \times n}$ ($n = |\mathcal{V}|$) for the current graph. The input is composed of a state feature $s_f$ and the information of node index $c$ for the current acting agent. Note that under partial observability, the global state $s$ is replaced by $(s_p, \text{Pos}, \text{belief})$. We use the shortest path distances to the $m$ pursuers as the initial feature of each node $v \in \mathcal{V}$. The normalized features of all $n$ nodes are concatenated with $(\text{Pos}, \text{belief})$ to construct the state feature $s_f$. Also note that the distances between one node and all other nodes can be computed using the $\mathcal{O}(n^2)$ Dijkstra algorithm.

Given the state feature input $s_f$, we embed it into $\mathbb{R}^{d \times n}$ and send the result into an encoder composed of 6 self-attention layers, where $d$ is the embedding dimension. Each layer takes the output $h$ of the last layer as the input and outputs $h'$ using a masked attention, whose time complexity is also $\mathcal{O}(n^2)$:

$$q_i = W_Q h_i, k_i = W_K h_i, \ v_i = W_V h_i, u_{ij} = \frac{q_i^T k_j}{\sqrt{d}}, w_{ij} = \frac{e^{u_{ij}}}{\sum_{t=1}^{n} e^{u_{it}}}, h_i' = \sum_{j=1}^{n} \min \left\{ w_{ij}, M_{ij} \right\} v_j,$$

where $W_Q, W_K, W_V \in \mathbb{R}^{d \times d}$ are the weights to be learned.

Given the output of the encoder $\hat{h}$, we employ a decoder without masks to gather global information. The decoder uses $\hat{h}_c$ to query in the output features $\hat{h}$ of all nodes, with the keys equal to the values:

$$q = W_Q \hat{h}_c, k_i = W_K \hat{h}_i, v_i = W_V \hat{h}_i, u_j = \frac{q^T k_j}{\sqrt{d}}, w_j = \frac{e^{u_j}}{\sum_{t=1}^{n} e^{u_t}}, \tilde{h}_c = \sum_{j=1}^{n} w_j v_j.$$

The decoder output $\tilde{h}_c$ is further concatenated with $\hat{h}_c$ and projected into $\mathbb{R}^d$. Then, it is used as a query for a pointer network, which takes the features of the neighbor nodes $\hat{h}_{ne}$ for the current agent as the keys and values. The pointer network directly outputs the attention vector $w$ as the current policy $\pi(\cdot | s)$ since the number of the neighbors aligns with the number of the valid actions. After the first query through the policy network, an action $a_1$ for the first agent is sampled from $\pi(\cdot | s)$, and the state is updated as $s' = (s, a_1)$. The subsequent $m - 1$ queries follow the same process as above.

## C.3 HYPERPARAMETER SETTING

Table 5 shows the detailed hyperparameter setting used in the training of our RL pursuer policy.

Table 5: Hyperparameter Setting of R2PS Training

| | |
|---|---|
| Discount factor $\gamma$ | 0.99 |
| SAC target entropy coefficient | 0.05 |
| GNN embedding dimension $d$ | 128 |
| GNN attention heads | 8 |
| Batch size | 128 |
| Learning rate | $10^{-5}$ |
| Update epoch | 8 |

## C.4 LEARNING CURVES OF RL PURSUERS

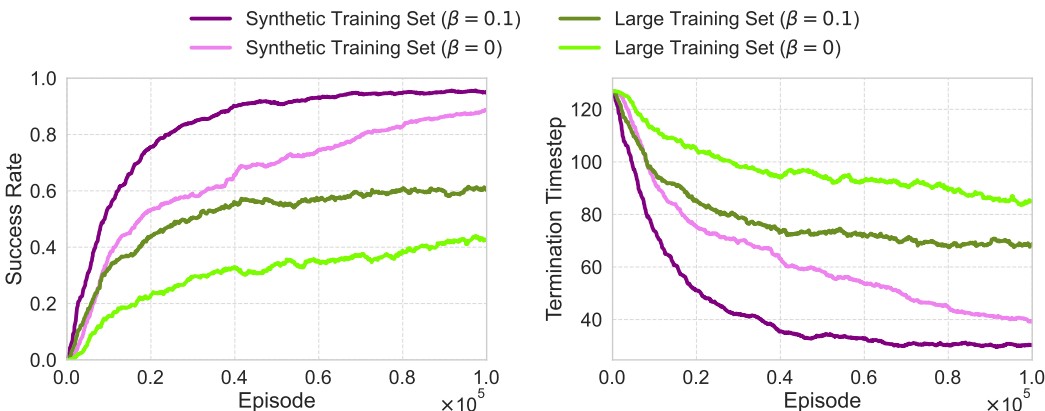

Figure 4: Cross-Graph Learning Curves of Generalized Pursuer Policies

During the cross-graph R2PS training, we consider the use of $\beta = 0.1$ and $\beta = 0$ in the policy loss $\mathcal{L}(\theta)$ (8). For the former, we employ the belief-averaged DP policy (6) as the reference policy. For the latter, it means that the training process is without policy guidance. Figure 4 shows the learning curves of our RL pursuer policies. Clearly, training with policy guidance is more efficient than pure reinforcement learning under SAC loss. This comparison verifies that the DP pursuer policy can serve as guidance to facilitate efficient exploration of the cross-graph RL policy. Besides, training under the synthetic training set is relatively easier than under the large one that contains more real-world graph structures. Nevertheless, our R2PS learning scheme gradually improves the quality of the RL pursuer policies under all of the four settings, using a very limited observation range of 2.

# D    EXPERIMENTAL DETAILS

## D.1    DETAILS OF TEST GRAPHS

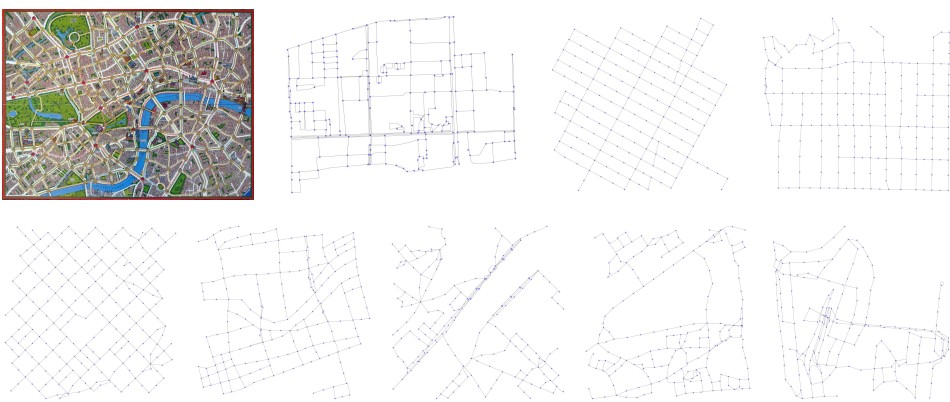

Figure 5: Illustration of Test Graphs (Starting from Scotland-Yard Map)

The test graphs for both DP and RL pursuers include Grid Map (a $10 \times 10$ grid), Scotland-Yard Map (from the board game Scotland-Yard), Downtown Map (a real-world location from Google Maps), and 7 famous real-world spots (from Times Square to Sydney Opera House). The graph structures are illustrated in Figure 5, following the order in Table 1.

The real-world graphs are generated through a program designed for discretizing the regions around the location centers in Google Maps. The map range is set to be 600 by default (corresponding to a radius of 600 meters). Nodes correspond to actual road intersections, endpoints, and geometry-defined shape points along roadways. Edges represent the physical road segments connecting these nodes, typically encoded as polylines that capture the true geometry of each street. To ensure a topologically coherent intersection structure, closely spaced intersection points (within 20 meters of one another) are abstracted into single representative nodes. After establishing the intersection-level skeleton, the program further adjusts the spatial resolution by subdividing any road segment whose length exceeds 100 (discretization granularity) meters. For such long segments, the program inserts additional intermediate points at regular intervals along the original road geometry. These newly added points are treated as supplementary nodes, and the original long segment is replaced by several shorter segments. The resulting graph therefore adopts a hybrid granularity: true intersections are preserved as primary nodes, while long road segments are discretized into shorter units of no more than 100 (discretization granularity) meters.

## D.2    ADDITIONAL RESULTS

Table 6: Success Rates of Belief-Averaged DP Pursuers under Different Observation Ranges

| Observation Range | 2 | 3 | 4 | 5 | 6 |
|---|---|---|---|---|---|
| Grid Map | 0.78 | 0.92 | 0.99 | 1.00 | 1.00 |
| Scotland-Yard Map | 0.63 | 0.95 | 1.00 | 1.00 | 1.00 |
| Downtown Map | 0.90 | 1.00 | 1.00 | 1.00 | 1.00 |
| Times Square, New York | 0.69 | 0.88 | 1.00 | 1.00 | 1.00 |
| Hollywood Walk of Fame, LA | 0.48 | 0.79 | 0.94 | 0.98 | 1.00 |
| Sagrada Familia, Barcelona | 0.36 | 0.70 | 0.92 | 0.96 | 1.00 |
| The Bund, Shanghai | 0.57 | 0.87 | 0.97 | 0.99 | 1.00 |
| Eiffel Tower, Paris | 0.94 | 0.98 | 0.99 | 1.00 | 1.00 |
| Big Ben, London | 0.74 | 0.94 | 1.00 | 1.00 | 1.00 |
| Sydney Opera House, Sydney | 0.87 | 0.96 | 0.99 | 0.99 | 1.00 |

Table 7: Success Rates of RL Pursuers under Different Observation Ranges

| Observation Range | 2 | 3 | 4 | 5 | 6 |
|---|---|---|---|---|---|
| Grid Map | 1.00 | 1.00 | 1.00 | 1.00 | 1.00 |
| Scotland-Yard Map | 0.76 | 0.98 | 0.99 | 0.99 | 1.00 |
| Downtown Map | 0.99 | 0.99 | 1.00 | 1.00 | 1.00 |
| Times Square, New York | 0.95 | 0.98 | 1.00 | 1.00 | 1.00 |
| Hollywood Walk of Fame, LA | 0.38 | 0.59 | 0.96 | 1.00 | 1.00 |
| Sagrada Familia, Barcelona | 0.20 | 0.72 | 0.88 | 0.95 | 0.96 |
| The Bund, Shanghai | 0.25 | 0.55 | 0.82 | 0.82 | 0.83 |
| Eiffel Tower, Paris | 1.00 | 1.00 | 1.00 | 1.00 | 1.00 |
| Big Ben, London | 0.82 | 0.95 | 0.98 | 0.99 | 0.99 |
| Sydney Opera House, Sydney | 0.95 | 0.98 | 1.00 | 1.00 | 1.00 |

We may find that Hollywood Walk of Fame, Sagrada Familia, and The Bund are relatively more difficult for the pursuers, especially under small observation ranges. Based on the statistics of the test graphs in Table 1 (left), here we provide a rough analysis of this phenomenon. In planar graphs, a large average degree generally implies the existence of small cycles. For example, in Grid Map, all minimal cycles' length is only $4$. Since successful evasions benefit more from large cycles, graphs like Grid Map, Scotland-Yard Map, and Downtown Map are easier for pursuit. Besides, Eiffel Tower, Big Ben, and Sydney Opera House all have large diameters, which implies the existence of long "links" that have poor connectivity with other nodes (see the last three graphs in Figure 5). Therefore, these graphs also benefit pursuit rather than evasion. As Hollywood Walk of Fame, Sagrada Familia, and The Bund do not have the mentioned characteristics, these graphs are harder for the pursuers.

Figure 6 provides the scaling plots of the computation (inference) time of DP and RL policies under an NVIDIA GeForce RTX 2080 Ti GPU, with the log-log plots on the right. Clearly, the time of DP computations significantly increases with the graph sizes. In comparison, the inference time of our GNN-based RL policy is only slightly longer in large graphs than in small graphs.

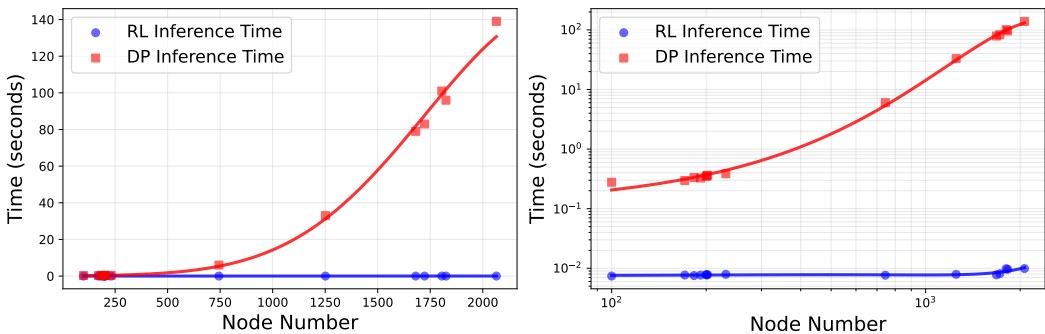

Figure 6: Scaling Plots of RL and DP Inference Time

## D.3 RL PERFORMANCE UNDER MORE PURSUERS

Table 8: Success Rates of RL Policy under Different Pursuer Numbers

| Pursuer Number | $m = 2$ | $m = 4$ | $m = 6$ |
|---|---|---|---|
| Grid Map | 1.00 | 1.00 | 1.00 |
| Scotland-Yard Map | 0.76 | 0.99 | 1.00 |
| Downtown Map | 0.99 | 1.00 | 1.00 |
| Times Square, New York | 0.95 | 0.97 | 1.00 |
| Hollywood Walk of Fame, LA | 0.38 | 0.82 | 0.93 |
| Sagrada Familia, Barcelona | 0.20 | 0.74 | 0.94 |
| The Bund, Shanghai | 0.25 | 0.99 | 1.00 |
| Eiffel Tower, Paris | 1.00 | 1.00 | 1.00 |
| Big Ben, London | 0.82 | 1.00 | 1.00 |
| Sydney Opera House, Sydney | 0.95 | 0.99 | 1.00 |

As our R2PS training is established upon the framework of EPG (Lu et al., 2025a), we can also employ the grouping mechanism proposed by Lu et al. (2025a) to derive pursuer and evader policies when the pursuer number $m$ is large. Table 8 compares the pursuit success rates of the multi-agent RL policies against the asynchronous-move DP evader under different pursuer numbers $m$. Clearly, the 4-pursuer policy can significantly increase the original success rates for $m = 2$. When $m = 6$, the success rates are close to 1 even under the fixed observation range of 2. Figure 7 further illustrates the FLOPs of our RL inference under different graph sizes and pursuer numbers.

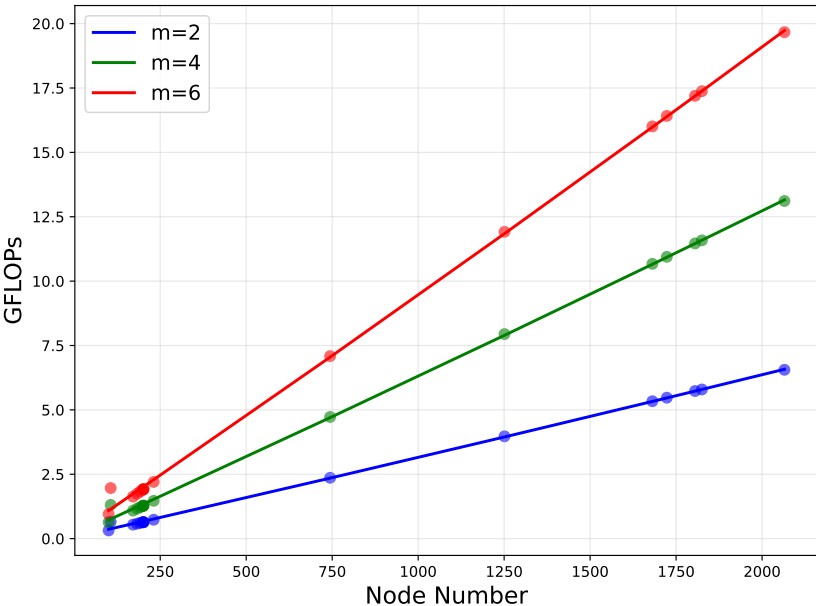

Figure 7: Scaling Plots of Floating-Point Operations under Different RL Pursuer Numbers

# E  RELATED WORK

**Finding optimal strategies in PEGs.** Graph-based pursuit-evasion games (PEGs) can be categorized into no-exit PEGs and multi-exit PEGs. The former is the primary form of pursuit-evasion and also the focus of this paper, while the latter is sometimes referred to as network security games. For no-exit PEGs, the early theoretical work (Goldstein & Reingold, 1995) proves that it requires exponential time to determine whether $m$ pursuers are sufficient to capture one evader on a given graph under perfect information. Vieira et al. (2008) provide the provably optimal method for solving sequential PEGs, and Chung et al. (2011) show that the basic idea behind PEG solving can be represented by a marking algorithm featuring state expansion. Horák & Bošanský (2017) consider the case where the pursuers only have partial observation and provide a dynamic programming algorithm in the form of value iteration for finding Nash equilibrium. Lu et al. (2025a) show that an expansion-based dynamic programming algorithm can solve Markov PEGs under a near-optimal time complexity. Recent works (Xue et al., 2021; 2022) combine neural networks with fictitious self-play and Monte-Carlo tree search to construct scalable deep reinforcement learning (RL) algorithms for finding robust pursuit strategies in network security games.

**Policy generalization in PEGs.** Policy-Space Response Oracles (PSRO) (Lanctot et al., 2017) is a standard game RL paradigm extended from the game-theoretic approach of double oracle (DO) (McMahan et al., 2003) for robust policy learning. While the approach itself is general, it can only solve PEGs on a designated graph structure, just like the methods above. As we have mentioned, policy generalization is crucial to real-time applications under real-world PEGs, becoming a focus of recent research. MT-PSRO (Li et al., 2023) combines multi-task policy pre-training with PSRO fine-tuning to enable few-shot generalization to unseen real-world opponents. Grasper (Li et al., 2024) proposes a two-stage pre-training method to facilitate few-shot generalization to unseen initial conditions of the game. Equilibrium Policy Generalization (EPG) (Lu et al., 2025a) provides a fundamentally novel paradigm to learn generalized policies across the underlying structures of PEGs through the construction of equilibrium oracles, guaranteeing robust zero-shot generalization to unseen graph structures in Markov PEGs. As EPG does not require the time-consuming PSRO tuning, the pursuit strategies are real-time applicable under full observability. However, as is mentioned in Lu et al. (2025a), whether such a kind of generalization can be applied to the case of imperfect information remains unclear. Since partial observability leads to the inherent PSPACE-hardness (see Papadimitriou & Tsitsiklis (1987)) and exponentially many information sets (see Lu et al. (2025b)), constructing equilibrium oracles directly under partial observability can be intractable.

