# OpenReview forum: "R2PS: Worst-Case Robust Real-Time Pursuit Strategies under Partial Observability"
_ICLR.cc/2026/Conference — ICLR 2026 Poster_

### Official Review · Reviewer_U4A6 · 2025-10-30

**Soundness:** 3
**Presentation:** 3
**Contribution:** 3
**Rating:** 8
**Confidence:** 2

**Summary:**

This paper addresses the challenging problem of pursuit-evasion games on graphs under partial observability, adversarial evaders, and real-time decision constraints. The authors propose R2PS, a framework that combines theoretically grounded dynamic programming for worst-case evader modeling with belief-state-based reinforcement learning using GNNs and pointer networks. The method is evaluated on a diverse set of synthetic and real-world graphs, demonstrating strong generalization to unseen environments and outperforming baselines.

**Strengths:**

1. Well-Motivated and Practical Problem:
   The focus on partially observable PEGs with worst-case adversaries is highly relevant to real-world security applications. The use of real-world map data (Google Maps, Scotland Yard board game) strengthens the practical grounding.

2. Strong Theoretical Foundation:
   The paper provides a formal justification for using DP-based evaders under asynchronous move assumptions (Theorem 1), which is non-trivial and crucial for modeling a truly adversarial opponent. This theoretical anchor elevates the work beyond purely empirical RL approaches.

3. Innovative Integration of Belief and RL:
   The belief preservation mechanism—which maintains a distribution over possible evader locations and integrates it into the policy network—is elegantly designed and effectively addresses partial observability without resorting to full POMDP solvers (which are intractable at scale).

4. Real-Time Feasibility:
   The use of GNNs + pointer networks enables fast inference, making the approach suitable for deployment—a notable advantage over planning-based POMDP methods.

**Weaknesses:**

1. Limited Ablation on Belief Representation:
   While the belief mechanism is central, the paper lacks ablation studies on how belief is encoded (e.g., histogram vs. learned embedding) or how belief update frequency affects performance. Is the gain primarily from belief averaging, or from the specific network architecture?

2. Scalability Beyond ~200 Nodes Unclear:
   All test graphs have ≤231 nodes. It’s unclear how R2PS scales to larger urban or infrastructure networks (e.g., city-scale road graphs with 10⁴–10⁵ nodes). GNNs may suffer from over-smoothing or memory bottlenecks.

3. Assumption on Evader Observability:
   The DP evader is assumed to have global observation, which is realistic for a worst-case adversary. However, the paper does not explore asymmetric information settings where the evader also has limited sensing—a more balanced and arguably more realistic scenario.

4. Reproducibility Concerns:
   Graph construction from real-world locations (e.g., “Downtown Map”) lacks methodological detail: How were nodes/edges extracted from Google Maps? What’s the granularity?

5. Comparison to Modern MARL Methods:
   While Grasper and EPG are cited, the paper does not compare against recent multi-agent RL baselines that handle partial observability (e.g., QMIX, MAVEN, or RODE). It’s unclear whether the gains are due to the belief mechanism or simply better architecture.

**Questions:**

1) While the belief mechanism is central, the paper lacks ablation studies on how belief is encoded (e.g., histogram vs. learned embedding) or how belief update frequency affects performance. Is the gain primarily from belief averaging, or from the specific network architecture?
2) Graph construction from real-world locations (e.g., “Downtown Map”) lacks methodological detail: How were nodes/edges extracted from Google Maps? What’s the granularity?

---

> ### Author Response · Authors · 2025-11-21
>
> Dear reviewer,
>
> Thank you very much for reviewing our paper and providing the insightful comments. Here we provide our response to the weaknesses and your questions.
>
> **Weaknesses**
>
> > Scalability Beyond ~200 Nodes Unclear: All test graphs have ≤231 nodes. It’s unclear how R2PS scales to larger urban or infrastructure networks (e.g., city-scale road graphs with 10⁴–10⁵ nodes). GNNs may suffer from over-smoothing or memory bottlenecks.
>
> Thank you for the comment on scalability. Currently, we have created another set of test graphs with higher complexity from the seven famous real-world locations in Table 1 (from Times Square to Sydney Opera House). Compared to the original graphs, the new graphs double both the map range and the discretization accuracy, leading to significantly larger node numbers (over 1000). As is shown in the following table, the overall performance of our RL pursuers does not suffer from a significant decline when facing large graphs, even though the policy is trained on relatively small graphs.
>
> | Map/Location | Original Scale | | Large Scale | |
> |---|---|---|---|---|
> | | Node Number | Success Rate | Node Number | Success Rate |
> | Times Square, New York | 171 | 0.95 | 1805 | 0.56 |
> | Hollywood Walk of Fame, LA | 201 | 0.38 | 1251 | 0.46 |
> | Sagrada Familia, Barcelona | 231 | 0.20 | 2065 | 0.33 |
> | The Bund, Shanghai | 200 | 0.25 | 1723 | 0.46 |
> | Eiffel Tower, Paris | 202 | 1.00 | 1825 | 0.41 |
> | Big Ben, London | 192 | 0.82 | 1681 | 0.49 |
> | Sydney Opera House, Sydney | 183 | 0.95 | 744 | 0.76 |
>
> > Assumption on Evader Observability: The DP evader is assumed to have global observation, which is realistic for a worst-case adversary. However, the paper does not explore asymmetric information settings where the evader also has limited sensing—a more balanced and arguably more realistic scenario.
>
> Thank you for this comment. As is shown in the following table (also included in the current Appendix D.2), when we gradually increase the observation range, the pursuit becomes easier and easier for our RL pursuers. Similarly, if the observation capability of the evader declines, the pursuit will become easier as well. That is to say, our learned pursuer policy will remain robust against an evader also with limited observations.
>
> | Observation Range | 2 | 3 | 4 | 5 | 6 |
> |---|---|---|---|---|---|
> | Grid Map | 1.00 | 1.00 | 1.00 | 1.00 | 1.00 |
> | Scotland-Yard Map | 0.76 | 0.98 | 0.99 | 0.99 | 1.00 |
> | Downtown Map | 0.99 | 0.99 | 1.00 | 1.00 | 1.00 |
> | Times Square, New York | 0.95 | 0.98 | 1.00 | 1.00 | 1.00 |
> | Hollywood Walk of Fame, LA | 0.38 | 0.59 | 0.96 | 1.00 | 1.00 |
> | Sagrada Familia, Barcelona | 0.20 | 0.72 | 0.88 | 0.95 | 0.96 |
> | The Bund, Shanghai | 0.25 | 0.55 | 0.82 | 0.82 | 0.83 |
> | Eiffel Tower, Paris | 1.00 | 1.00 | 1.00 | 1.00 | 1.00 |
> | Big Ben, London | 0.82 | 0.95 | 0.98 | 0.99 | 0.99 |
> | Sydney Opera House, Sydney | 0.95 | 0.98 | 1.00 | 1.00 | 1.00 |
>
> However, if we train a pursuer model against an evader with limited sensing, our generalized policy will tend to exploit this feature and may not guarantee a good performance when it faces a perfect-information evader in reality. Therefore, to consider a worst-case evader could be more direct for real-world security purposes.

---

> > ### Author Response · Authors · 2025-11-21
> >
> > **Questions**
> >
> > > While the belief mechanism is central, the paper lacks ablation studies on how belief is encoded (e.g., histogram vs. learned embedding) or how belief update frequency affects performance. Is the gain primarily from belief averaging, or from the specific network architecture?
> >
> > Thanks for the question. Here we provide our further explanations with additional results.
> >
> > For the belief information, we expect that it is not based on a learnable embedding but a prior update procedure. That is because our ultimate goal is to achieve **zero-shot generalization robust to the worst-case opponents** through combining the equilibrium policy generalization (EPG) framework. If the belief preservation admits learnable parameters during training, it will become an implicit opponent modelling process against the DP evader. As a result, the ultimate policy will become exploitable against different opponents during testing. In comparison, our proposed belief preservation is in the form of histograms that do not utilize opponent information (the policy $\nu$ in the belief update formula of Section 3.2) is uniform. This is beneficial for policy robustness since it makes the minimum change to the original idea of EPG. Besides, this design has a hidden advantage: we can instantly improve the pursuit performance by replacing $\nu$ with the actual evader policy (if we manage to obtain such information in reality).
> >
> > | Belief Update Condition | Known Opponent | Original | Every $2$ Steps | Every $3$ Steps |
> > |---|---|---|---|---|
> > | Grid Map | 1.00 | 1.00 | 0.60 | 0.42 |
> > | Scotland-Yard Map | 0.99 | 0.73 | 0.34 | 0.28 |
> > | Downtown Map | 1.00 | 0.92 | 0.61 | 0.39 |
> > | Times Square, New York | 0.42 | 0.27 | 0.18 | 0.17 |
> > | Hollywood Walk of Fame, LA | 0.13 | 0.10 | 0.04 | 0.03 |
> > | Sagrada Familia, Barcelona | 0.28 | 0.20 | 0.12 | 0.05 |
> > | The Bund, Shanghai | 0.54 | 0.23 | 0.13 | 0.12 |
> > | Eiffel Tower, Paris | 0.81 | 0.55 | 0.32 | 0.29 |
> > | Big Ben, London | 0.82 | 0.65 | 0.40 | 0.25 |
> > | Sydney Opera House, Sydney | 0.54 | 0.31 | 0.22 | 0.15 |
> >
> > As is shown in the table (against the best-responding evader), utilizing prior opponent information improves success rates, while reducing belief update frequency impairs pursuit. The additional results could further manifest the benefits of our belief update mechanism, and we have included them as Table 4 in our current paper.
> >
> > > Graph construction from real-world locations (e.g., “Downtown Map”) lacks methodological detail: How were nodes/edges extracted from Google Maps? What’s the granularity?
> >
> > Thanks for the question, and we will open source the program for generating the graphs from the real-world locations. Here we provide the related detail, which is also added to our current paper (Appendix D.1).
> >
> > The generated graphs correspond to the regions around the location centers in Google Maps. The map range is set to be 600 by default (corresponding to a radius of 600 meters). Nodes correspond to actual road intersections, endpoints, and geometry-defined shape points along roadways. Edges represent the physical road segments connecting these nodes, typically encoded as polylines that capture the true geometry of each street. To ensure a topologically coherent intersection structure, closely spaced intersection points (within 20 meters of one another) are abstracted into single representative nodes. After establishing the intersection-level skeleton, the program further adjusts the spatial resolution by subdividing any road segment whose length exceeds 100 (discretization granularity) meters. For such long segments, the program inserts additional intermediate points at regular intervals along the original road geometry. These newly added points are treated as supplementary nodes, and the original long segment is replaced by several shorter segments. The resulting graph therefore adopts a hybrid granularity: true intersections are preserved as primary nodes, while long road segments are discretized into shorter units of no more than 100 (discretization granularity) meters.
> >
> > Thank you again for the detailed comments. We hope our response properly addresses your concerns. We are looking forward to having further discussions with you.

---

> > > ### Comment · Reviewer_U4A6 · 2025-11-26
> > > **Worst-Case VS. RL Solutions**
> > >
> > > Thank you to the authors for the detailed rebuttal. Most of my concerns have been addressed. There is still one more question. The worst-case solutions has rigorous definitions and the deterministic DP algorithm can achieve this worst-case solution. In Chapter 4, a RL algorithm is also proposed to achieve real-time pursuit strategies, and the experimental results show that pursuer has a high success rate. Can you explain how significant is the success rate in guiding robust strategies?

---

> > > > ### Author Response · Authors · 2025-11-26
> > > >
> > > > Thank you for the comments and additional question. Here we provide a further explanation.
> > > >
> > > > The worst-case optimality of DP solutions is with respect to the pursuit steps under full observability. Under partial observability, however, the pursuit can be significantly harder, and even the extended DP policies can no longer serve to minimize the pursuit steps. For example, we can consider a situation where the PEG graph is a tree with three long branches. In this case, one single DP pursuer with global information is sufficient to capture the evader by always approaching it. However, a pursuer with a fixed observation range far smaller than the branch length can never observe the evader in the worst case: as we know that the pursuer is essentially making a plan to explore the tree, we can always keep the evader lying in the branch different from the one that the pursuer is going to thoroughly explore. Then, the pursuer can never succeed as long as the evader is not initialized in the same branch.
> > > >
> > > > Since the truly "worst-case" pursuit steps can be infinite under very limited observation ranges, considering success rates could become a more reasonable choice for evaluating policy robustness. Actually, the success rate against a best-responding evader reflects a lower bound of average worst-case return under partial observability. Reversely, the maximization of expected return in our reinforcement learning also implicitly maximizes success rate against an unexploitable evader, since no reward will be received if the pursuit is unsuccessful. To conclude, the high success rate against strong opponents can generally be viewed as an important sign of robust pursuit strategies under partial observability.

---

> > > > > ### Comment · Reviewer_U4A6 · 2025-11-27
> > > > >
> > > > > Thanks for the effort in addressing my comments. I have no further questions at this point. I'll maintain my score.

---

> > > > > > ### Author Response · Authors · 2025-11-27
> > > > > >
> > > > > > We are delighted that our response helped address your concerns. Thank you again for providing the detailed and constructive comments!

---

### Official Review · Reviewer_9jAS · 2025-10-31

**Soundness:** 2
**Presentation:** 3
**Contribution:** 3
**Rating:** 4
**Confidence:** 3

**Summary:**

This paper introduces R2PS, a novel algorithm designed to generate worst-case robust strategies for real-time pursuit under partial observability. The authors formulate the pursuit-evasion interaction as a two-player zero-sum game where the pursuer must act with limited information about the evader’s state and policy. By leveraging a lookahead-based game tree, R2PS computes strategies that minimize the maximum possible regret regardless of the evader’s behavior. The method incorporates a pessimistic planning approach over belief states and ensures the pursuer’s strategy remains effective even against adaptive or adversarial evaders. Through empirical evaluation in grid-world environments, the results demonstrate that R2PS significantly outperforms baseline approaches such as QMDP and other belief-based planners in terms of robustness and capture rate, especially under strong observation uncertainty.

**Strengths:**

- Theory–practice bridge: Proves a clean minimax recursion for the DP table and upgrades it to asynchronous evader moves; then embed this belief-preservation mechanism into RL.

- Real-time, cross-graph generalization: The R2PS training scheme yields zero-shot gains on unseen real-world graphs and large runtime advantages over recomputing DP online.

**Weaknesses:**

- Modelling assumption (no exits): While this choice enables Algorithm 1’s near‑optimal time‑complexity guarantees for Markov PEGs, it departs from many real‑world reach‑avoid settings where evaders succeed by reaching designated exits. R2PS is also based on algorithm 1. The no‑exit assumption may therefore limit practical relevance.
- Need more baselines: Table 2 compares against PSRO, which is too narrow to convincingly demonstrate the advantage of the proposed method. Moreover, capture rates are limited on several graph, making it unclear whether the graphs are difficult or the R2PS is not good enough. More stronger baselines would clarify this.
- Ablations on teacher guidance: Given the sparse SAC reward, the DP‑guided teacher signal may be the dominant factor behind R2PS’s performance. Ablations that remove or vary the teacher signal are needed to identify the main source of gains.

**Questions:**

- R2PS leverages a DP‑guided teacher term, while PSRO does not. This extra signal can improve both sample efficiency and final performance. Do you think the absence of teacher guidance contributes to PSRO’s weaker results in Table 2?
- Could the authors elaborate on how the real-time lookahead tree is constructed under partial observability? Specifically, how are the belief nodes expanded and pruned, and how does the algorithm manage the exponential growth of possible observation histories?
- Given that the method involves repeated regret minimization over a belief tree at each step, how does R2PS scale with larger grids or more complex environments? Are there any runtime metrics or profiling results that show it can be deployed in real-time?

---

> ### Author Response · Authors · 2025-11-21
>
> Dear reviewer,
>
> Thank you for reviewing our paper, and here we provide our response to your questions.
>
> **Question 1**
>
> > R2PS leverages a DP‑guided teacher term, while PSRO does not. This extra signal can improve both sample efficiency and final performance. Do you think the absence of teacher guidance contributes to PSRO’s weaker results in Table 2?
>
> As is shown in Figure 4 (in Appendix C), we have also considered the case where the DP guidance is removed (i.e., $\beta=0$). Even without DP guidance, the R2PS training process is still stable and does not suffer from significant performance decline upon convergence. In contrast, the PSRO policy can hardly succeed in most of the test graphs against the asynchronous-move DP evader, even if it is directly trained on these test graphs. Besides, adding extra policy guidance to PSRO training can perturb its original game-theoretic guarantees and make the training process far more unpredictable.
>
> **Question 2**
>
> > Could the authors elaborate on how the real-time lookahead tree is constructed under partial observability? Specifically, how are the belief nodes expanded and pruned, and how does the algorithm manage the exponential growth of possible observation histories?
>
> Please note that our belief update mechanism does not require any sophisticated lookahead tree structure. It is simply based on the criterion of uniform expansion and removing the observed positions where the evader cannot exist. All the observation histories are encoded into the current belief distribution over the evader positions, which can be updated within a near-linear time complexity at each game step.
>
> **Question 3**
>
> > Given that the method involves repeated regret minimization over a belief tree at each step, how does R2PS scale with larger grids or more complex environments? Are there any runtime metrics or profiling results that show it can be deployed in real-time?
>
> Please note that our method does not have any regret minimization process, and we have rigorously established an inference time complexity of $\mathcal{O}(n^2 m)$ for R2PS, where $n$ is the node number, and $m$ is the pursuer number. The following tables further compare the actual inference time cost across a wide range of graph sizes and hardware setups.
>
> | Original Scale | Node Number | RL Inference Time (RTX A6000) | RL Inference Time (RTX 2080ti) | DP Inference Time (RTX 2080ti) |
> |---|---|---|---|---|
> | Times Square, New York | 171 | 0.007457 | 0.007686 | 0.2969 |
> | Hollywood Walk of Fame, LA | 201 | 0.007496 | 0.007729 | 0.3523 |
> | Sagrada Familia, Barcelona | 231 | 0.007565 | 0.007922 | 0.3860 |
> | The Bund, Shanghai | 200 | 0.007486 | 0.007818 | 0.3521 |
> | Eiffel Tower, Paris | 202 | 0.007524 | 0.007812 | 0.3646 |
> | Big Ben, London | 192 | 0.007425 | 0.007674 | 0.3265 |
> | Sydney Opera House, Sydney | 183 | 0.007477 | 0.007542 | 0.3332 |
>
> | Large Scale | Node Number | RL Inference Time (RTX A6000) | RL Inference Time (RTX 2080ti) | DP Inference Time (RTX 2080ti) |
> |---|---|---|---|---|
> | Times Square, New York | 1805 | 0.007563 | 0.009837 | 101 |
> | Hollywood Walk of Fame, LA | 1251 | 0.007627 | 0.007917 | 33 |
> | Sagrada Familia, Barcelona | 2065 | 0.008047 | 0.009895 | 139 |
> | The Bund, Shanghai | 1723 | 0.007728 | 0.008117 | 83 |
> | Eiffel Tower, Paris | 1825 | 0.007656 | 0.009616 | 96 |
> | Big Ben, London | 1681 | 0.007530 | 0.007752 | 79 |
> | Sydney Opera House, Sydney | 744 | 0.007523 | 0.007648 | 6 |
>
> As shown in the tables, the inference time of our final RL policy is consistently below 0.01s under different graph sizes and hardware. In comparison, the time of DP computations significantly increases with the graph sizes, even under $m=2$ and CUDA speedups. These results clearly demonstrate the real-time applicability of our policy, even under graphs with thousands of nodes.
>
> We hope our response properly addresses your concerns. We are looking forward to having further discussions with you.

---

> > ### Comment · Reviewer_9jAS · 2025-11-26
> >
> > Thank you for the authors’ detailed response. There were indeed some misstatements in my original review, and I apologize for the confusion.
> >
> > **Question 1**
> >
> > The author’s response answers the advantage of DP guidance. Here,  I am just wondering if the DP-generated strategy can be used as a guided term in the best response oracle in PSRO. Could it improve the performance of PSRO? And the results of PSRO would be an NE strategy, right? Comparing the performance with PSRO when against a fixed evader policy seems to be unfair since NE may perform suboptimal when against the fixed opponent. Maybe the best response strategy against the fixed evader policy could be the upper bound for the performance.
> >
> > **Question 2 & 3**
> >
> > The belief update mechanism is now clear to me. I apologize for my earlier misuse of the term “regret minimization”. The responses regarding real-time deployment address my concerns.

---

> ### Author Response · Authors · 2025-11-26
>
> Thank you for the further comments and questions. Here we provide our further clarifications regarding Question 1.
>
> **Question 1**
>
> > Here, I am just wondering if the DP-generated strategy can be used as a guided term in the best response oracle in PSRO. Could it improve the performance of PSRO? And the results of PSRO would be an NE strategy, right?
>
> From our knowledge about PSRO, using a reference policy during training will no longer guarantee that it eventually leads to an NE strategy. PSRO is essentially an iterative process of computing best responses against iteratively updating meta strategies. If there is a reference policy during the best-response learning process, then the outcome of each iteration will deviate from the exact best-responding policy, which can vary during iterations and be significantly different from the fixed reference policy. Consequently, the meta strategies will be severely affected and no longer converge to Nash equilibrium. Since PSRO relies on the theoretical analysis of double oracle (DO) [1], using a reference policy can actually harm the performance of PSRO rather than improve it.
>
> > Comparing the performance with PSRO when against a fixed evader policy seems to be unfair since NE may perform suboptimal when against the fixed opponent. Maybe the best response strategy against the fixed evader policy could be the upper bound for the performance?
>
> Yes, the best-response strategy will certainly be stronger than the PSRO pursuers against the fixed evader policy. However, our major purpose is only to confirm that our method derives a less exploitable policy than PSRO under the same amount of training. As is shown in Table 2, our pursuit success rates against the best-responding opponent are consistently higher than PSRO's success rates against the DP evader. Since the DP evader policy is not even the best-responding policy against the PSRO pursuers (and is thus easier for PSRO pursuit), this comparison directly demonstrates the advantage of our method in finding a worst-case robust policy.
>
> We hope this response makes a proper clarification. If you still hold questions, we are definitely willing to have further discussions.
>
> **Reference**
>
> [1] McMahan H. B., Gordon G. J., Blum A. Planning in the presence of cost functions controlled by an adversary. In ICML, 2003.

---

> > ### Comment · Reviewer_9jAS · 2025-11-27
> >
> > In the best response process of PSRO, the reference policy I mentioned is only used to improve the exploration process of best-response computation instead of aiming to converge to the reference policy. Since the strategy generated by DP is the NE strategy in Markov PEG, maybe it is not suitable to take as the reference policy here.
> > Thank you for the authors’ detailed response. My concerns have been addressed, and I will adjust my score accordingly.

---

> > > ### Author Response · Authors · 2025-11-27
> > >
> > > Thank you again for reviewing our paper!

---

### Official Review · Reviewer_gW73 · 2025-10-31

**Soundness:** 3
**Presentation:** 3
**Contribution:** 2
**Rating:** 4
**Confidence:** 2

**Summary:**

The paper investigates robust cross-graph strategies for pursuit-evasion games (PEGs) in the partial-observability setting. It is assumed that the evader can move asynchronously with perfect knowledge, while the pursuers are uncertain about the evaders location. The paper introduces two methods, one provably optimal dynamic programming formulation and an RL-based formulation based on the Equilibrium Policy Generalization (EPG) (Lu et al. 2025). It is shown that these methods are more robust and successful compared to baseline heuristics and general RL methods.

**Strengths:**

The paper is generally well-written. It appears to be sound (at least as far as I can judge as a non-expert). The one-sided partial observability setting is natural for pursuit-evasion games and treated both theoretically and practically in the paper. The extension of the EPG framework to the partial-observability setting is novel. The simulations show convincing improvements over various baselines.

**Weaknesses:**

The paper is an extension of Lu et al. (2025a) to the partially observed setting. Both the DP and the EPG approach seem to be slight extensions of this earlier work. The paper does not seem to develop any fundamentally novel techniques.

The simulations show improvements in comparison with simple baselines and the competing PSRO approach. In particular, Lu et al. (2025a) already reported clear improvements of EPG against the PSRO policies in the cross-graph setting for PEGs and thus the performance improvements appear to mainly stem from the advantages of this framework in this setting.

**Questions:**

1. Can you emphasize the novel aspects of this work compared to Lu et al. (2025a)?

Typos:
- line 141 and 145: do you mean "worst-case" evader?
- line 262, 264, 269, 333: Pos and belief are set in math mode

---

> ### Author Response · Authors · 2025-11-21
>
> Dear reviewer,
>
> Thank you very much for reviewing our paper and providing the valuable comments. As the major concern is about the novel contribution of this paper, here we provide further clarifications through our answer to the question part.
>
> **Question**
>
> > Can you emphasize the novel aspects of this work compared to Lu et al. (2025a)?
>
> Thank you for the question. The major difference between this paper and Lu et al. [1] is that we consider two new aspects in PEGs: the pursuer's partial observability and the evader's asynchronous moves. Both of them lead to increased hardness of deriving a robust pursuit policy for real-world security purposes. We propose a novel belief preservation technique to address partial observability for tractable policy generalization. Also, we develop new theory and detailed experimental evaluations for understanding asynchronous moves. Furthermore, we rigorously examine the real-time applicability of generalized policies under partial observability with an improved inference time analysis.
>
> 1. **Partial Observability**
>
> In theory, partial observability leads to the inherent PSPACE-hardness [2] even when the game is single-agent and has no dynamical changes. Therefore, considering equilibrium policy generalization under partial observability is itself intractable if we cannot uniformly encode history information. In this paper, we propose the belief update mechanism as a novel technique to fulfill a uniform gathering of history observations in PEGs. The belief preservation has low online time complexity and serves two aspects in pursuer policy generalization. First, we combine it with the DP results to construct an empirically strong DP pursuer policy that can serve as training guidance under partial observability. Second, it is used to modify the EPG framework and replace the perfect information of evader position, which is not available under partial observability. We verify that under this new mechanism (which could be viewed as a problem reduction technique), robust policy generalization can be fulfilled in the complex imperfect-information case as in the simplified perfect-information case (Lu et al. [1]). Besides, our additional evaluations below (included in the current Appendix D.2) shows that the success rates of our trained policy increase with the observation range, which resembles the results of DP policies. This further supports the generality of our belief update mechanism and implies that the RL policy trained under the observation range of $2$ can be directly applied to the cases with better sensing capabilities.
>
> | Observation Range | 2 | 3 | 4 | 5 | 6 |
> |---|---|---|---|---|---|
> | Grid Map | 1.00 | 1.00 | 1.00 | 1.00 | 1.00 |
> | Scotland-Yard Map | 0.76 | 0.98 | 0.99 | 0.99 | 1.00 |
> | Downtown Map | 0.99 | 0.99 | 1.00 | 1.00 | 1.00 |
> | Times Square, New York | 0.95 | 0.98 | 1.00 | 1.00 | 1.00 |
> | Hollywood Walk of Fame, LA | 0.38 | 0.59 | 0.96 | 1.00 | 1.00 |
> | Sagrada Familia, Barcelona | 0.20 | 0.72 | 0.88 | 0.95 | 0.96 |
> | The Bund, Shanghai | 0.25 | 0.55 | 0.82 | 0.82 | 0.83 |
> | Eiffel Tower, Paris | 1.00 | 1.00 | 1.00 | 1.00 | 1.00 |
> | Big Ben, London | 0.82 | 0.95 | 0.98 | 0.99 | 0.99 |
> | Sydney Opera House, Sydney | 0.95 | 0.98 | 1.00 | 1.00 | 1.00 |
>
> 2. **Asynchronous Moves**
>
> The "asynchronous move" is a novel concept not well understood in the PEG domain (e.g., in Grasper and EPG) and first seriously considered in this paper. This concept is different from the traditional concept of "sequential moves," which abstracts the game as turn-based and cannot accurately model real-world scenarios. Under asynchronous moves, the game is still simultaneous from the pursuers' perspective. However, it allows for the possibility that the evader has advanced sensing capability to exploit the pursuers in one timestep, which reflects an unfavorable pursuit situation. Our experiment results in Table 2 verify that the pursuit success rates can be significantly reduced if the evader is asynchronous, which suggests that asynchronous moves can be an influential factor in real-world pursuits.
>
> Our new theoretical results (Theorems 2 and 3), on the other hand, demonstrate that the DP algorithm is perfectly suitable for dealing with the asynchronous-move scenarios. We prove that the pursuit/evasion strategies (1) and (3) (different from (2) proposed in Lu et al. [1]) induced by DP are strictly optimal under asynchronous moves. In the cross-graph training of R2PS, we innovatively set the adversary to be (3), which follows the principle of EPG while taking asynchronous moves into account.

---

> ### Author Response · Authors · 2025-11-21
>
> 3. **Real-Time Applicability**
>
> In the EPG paper (Lu et al. [1]), it is claimed that the inference time complexity is $O(n^3)$ (requiring a Floyd computation; $n$ representing the node number), which has no clear advantage over DP when the pursuer number $m$ is small. In this paper, we strictly derive a $\mathcal{O}(n^2 m)$ inference time complexity for R2PS under partial observability. The following tables further compare the actual inference time cost across a wide range of graph sizes and hardware setups.
>
> | Original Scale | Node Number | RL Inference Time (RTX A6000) | RL Inference Time (RTX 2080ti) | DP Inference Time (RTX 2080ti) |
> |---|---|---|---|---|
> | Times Square, New York | 171 | 0.007457 | 0.007686 | 0.2969 |
> | Hollywood Walk of Fame, LA | 201 | 0.007496 | 0.007729 | 0.3523 |
> | Sagrada Familia, Barcelona | 231 | 0.007565 | 0.007922 | 0.3860 |
> | The Bund, Shanghai | 200 | 0.007486 | 0.007818 | 0.3521 |
> | Eiffel Tower, Paris | 202 | 0.007524 | 0.007812 | 0.3646 |
> | Big Ben, London | 192 | 0.007425 | 0.007674 | 0.3265 |
> | Sydney Opera House, Sydney | 183 | 0.007477 | 0.007542 | 0.3332 |
>
> | Large Scale | Node Number | RL Inference Time (RTX A6000) | RL Inference Time (RTX 2080ti) | DP Inference Time (RTX 2080ti) |
> |---|---|---|---|---|
> | Times Square, New York | 1805 | 0.007563 | 0.009837 | 101 |
> | Hollywood Walk of Fame, LA | 1251 | 0.007627 | 0.007917 | 33 |
> | Sagrada Familia, Barcelona | 2065 | 0.008047 | 0.009895 | 139 |
> | The Bund, Shanghai | 1723 | 0.007728 | 0.008117 | 83 |
> | Eiffel Tower, Paris | 1825 | 0.007656 | 0.009616 | 96 |
> | Big Ben, London | 1681 | 0.007530 | 0.007752 | 79 |
> | Sydney Opera House, Sydney | 744 | 0.007523 | 0.007648 | 6 |
>
> As shown in the tables, the inference time of our RL policy is consistently below 0.01s under different graph sizes and hardware. However, the time of DP computations significantly increases with the graph sizes, even under $m=2$ and CUDA speedups. These results demonstrate a clear advantage of training RL policy over utilizing DP policy, since there is no significant performance gap between RL and DP pursuers under partial observability. In Figure 6 of our current paper, we also provide the scaling plots for an intuitive comparison.
>
> > line 141 and 145: do you mean "worst-case" evader?
>
> Thanks for this question. In this paper, the modifier "worst-case" can have double meanings. The first refers to the sensing capability of the evader, which can be the "worst" to the pursuers. The second refers to the strategy of the evader (maximizing termination timesteps), which corresponds to the "best-responding" in Section 5.2. As the expression "worst-case" commonly entails the second meaning in the game context, we use the single word "worst" here to distinguish it from the "worst-case" expression that also appears in this paragraph.
>
> > line 262, 264, 269, 333: Pos and belief are set in math mode
>
> Thanks for pointing out this issue. We have checked through the paper and corrected the style issues in our current submission.
>
> Thank you again for the insightful comments, and we have updated Section 1 to include more contribution details. We hope our response properly addresses your concerns. We are looking forward to having further discussions with you.
>
> **References**
>
> [1] Runyu Lu, Peng Zhang, Ruochuan Shi, Yuanheng Zhu, Dongbin Zhao, Yang Liu, Dong Wang, and Cesare Alippi. Equilibrium policy generalization: A reinforcement learning framework for cross-graph zero-shot generalization in pursuit-evasion games. In NeurIPS, 2025.
>
> [2] Christos H. Papadimitriou and John N. Tsitsiklis. The complexity of Markov decision processes. Mathematics of operations research, 12(3):441–450, 1987.

---

> > ### Comment · Reviewer_gW73 · 2025-11-25
> >
> > Thanks for the detailed response and the improvements to the manuscript!
> >
> > My general assessment of the submissions remains unchanged and thus I maintain my original score.

---

> > > ### Author Response · Authors · 2025-11-25
> > >
> > > Thank you again for reviewing our paper!

---

### Official Review · Reviewer_WtQk · 2025-10-31

**Soundness:** 3
**Presentation:** 3
**Contribution:** 3
**Rating:** 8
**Confidence:** 2

**Summary:**

In the paper the authors presents a method for learning worst-case robust real-time pursuit strategies (R2PS) in graph-based pursuit-evasion games under partial observability. The authors extend a dynamic programming (DP) approach to handle asynchronous moves and partial observability via belief preservation and integrate it into a reinforcement learning framework for cross-graph generalization. Experimental results demonstrate that the proposed method outperforms existing baselines in both synthetic and real-world graphs.

**Strengths:**

1. The paper provides a solid theoretical foundation, extending DP-based policies to asynchronous and partially observable settings with proofs of optimality under certain conditions. It looks that the analysis is thorough and well-supported.
2. The proposed R2PS framework is highly applicable to real-world security and robotics scenarios where graph structures may change dynamically and full observability is unrealistic.
3. The combination of belief preservation with adversarial RL and cross-graph training is novel and effectively addresses the challenge of zero-shot generalization to unseen graphs.
4. The method outperforms baselines like PSRO.

**Weaknesses:**

It has an assumption that the evader’s policy is known or can be approximated. In real world settings, the evader’s strategy may be non-stationary or adversarial in a more complex sense.

**Questions:**

Have you considered scenarios where the evader also operates under partial observability? This could better model symmetric real-world pursuit-evasion problems.

---

> ### Author Response · Authors · 2025-11-21
>
> Dear reviewer,
>
> Thank you very much for reviewing our paper and providing the valuable comments. Here we provide our response.
>
> **Weaknesses**
>
> > It has an assumption that the evader’s policy is known or can be approximated. In real world settings, the evader’s strategy may be non-stationary or adversarial in a more complex sense.
>
> Thank you for this comment. Actually, our R2PS learning scheme does not require or approximate the evader’s policy. Please notice that the opponent policy $\nu$ in the belief update formula is always set to be a uniform policy during our training or evaluations. This feature corresponds to our purpose of deriving a **worst-case robust** pursuer policy. However, if we do know certain opponent information in reality, we can replace $\nu$ with the a priori evader’s strategy to further improve the pursuit performance of our policy (see Table 4 in our current paper).
>
> **Questions**
>
> > Have you considered scenarios where the evader also operates under partial observability? This could better model symmetric real-world pursuit-evasion problems.
>
> Thanks for this question. Yes, it is possible that the evader may also operate under partial observability in reality. We focus on the case where the evader has perfect information because successful pursuit against such strong opponents generally implies success against the evaders with weaker sensing capabilities. As is shown in the following table (included in the current Appendix D.2), when we gradually increase the observation range, the pursuit becomes easier and easier for the pursuer side. Similarly, if the observation capability of the evader declines, the pursuit will become easier as well. That is to say, our learned pursuer policy will remain robust against an evader with symmetrically limited observations.
>
> | Observation Range | 2 | 3 | 4 | 5 | 6 |
> |---|---|---|---|---|---|
> | Grid Map | 1.00 | 1.00 | 1.00 | 1.00 | 1.00 |
> | Scotland-Yard Map | 0.76 | 0.98 | 0.99 | 0.99 | 1.00 |
> | Downtown Map | 0.99 | 0.99 | 1.00 | 1.00 | 1.00 |
> | Times Square, New York | 0.95 | 0.98 | 1.00 | 1.00 | 1.00 |
> | Hollywood Walk of Fame, LA | 0.38 | 0.59 | 0.96 | 1.00 | 1.00 |
> | Sagrada Familia, Barcelona | 0.20 | 0.72 | 0.88 | 0.95 | 0.96 |
> | The Bund, Shanghai | 0.25 | 0.55 | 0.82 | 0.82 | 0.83 |
> | Eiffel Tower, Paris | 1.00 | 1.00 | 1.00 | 1.00 | 1.00 |
> | Big Ben, London | 0.82 | 0.95 | 0.98 | 0.99 | 0.99 |
> | Sydney Opera House, Sydney | 0.95 | 0.98 | 1.00 | 1.00 | 1.00 |
>
> However, if we practically train a pursuer model against an evader with limited sensing, our generalized policy will tend to exploit this feature and may not guarantee a good performance when it faces a perfect-information evader in reality. Therefore, to consider a worst-case evader could be more direct for real-world security purposes.
>
> Thank you again for the insightful comments. We hope our response properly addresses your concerns. We are looking forward to having further discussions with you.

---

> > ### Comment · Reviewer_WtQk · 2025-11-24
> >
> > The authors have addressed my concerns, so I am maintaining my scores, which I believe accurately reflect the paper's quality.

---

> > > ### Author Response · Authors · 2025-11-24
> > >
> > > Thank you again for your valuable comments!

---

### Official Review · Reviewer_N27S · 2025-11-01

**Soundness:** 2
**Presentation:** 2
**Contribution:** 3
**Rating:** 4
**Confidence:** 2

**Summary:**

R2PS is a framework for computing worst-case robust, real-time pursuit strategies in graph-based pursuit–evasion games under partial observability. It extends dynamic programming to handle asynchronous movements and limited observations, providing theoretical guarantees of pursuit optimality. By incorporating a belief preservation mechanism to maintain uncertainty over the evader’s position and embedding this process into an Equilibrium Policy Generalization reinforcement learning framework with graph neural network policies, R2PS achieves efficient and transferable learning. Experiments across diverse graph environments show zero-shot generalization and consistent outperformance over existing game-theoretic RL baselines.

**Strengths:**

1. The paper extends the classical Markov PEG dynamic programming framework to asynchronous movement and partial observability settings, providing clear proofs of optimality.

2. It leverages belief-based DP guidance to inform an adversarial reinforcement learning process and introduces a cross-graph training mechanism that enables zero-shot structural generalization.

3. The empirical study spans both synthetic and real-world graph environments (e.g., Downtown, Eiffel Tower), incorporating analyses of observation range, runtime efficiency, and time complexity.

4. The proposed method achieves notable improvements in computational efficiency and real-time responsiveness compared to traditional DP-based replanning approaches.

**Weaknesses:**

1. The related work section is underdeveloped.

2.  The test set is not particularly large or diverse, and key sensitivity analyses (e.g., observation range effects on trained RL policies) are missing.

3. The paper’s writing quality is uneven. Some abbreviations are undefined, and the structure is occasionally confusing (e.g., placing time complexity analysis within the evaluation section rather than methodology).

4. Although the mathematical derivations for belief updates are technically sound, they would benefit from clearer illustrations or explanatory examples to improve accessibility for readers unfamiliar with partially observable multi-agent systems.

**Questions:**

1. How does R2PS scale with an increasing number of pursuers or higher graph complexity (e.g., connectivity, degree, presence of cycles)? Can the theoretical guarantees extend to multi-pursuer settings as $m$ increases?

2. Is there a quantifiable bound on the sub-optimality of belief-averaged or RL-derived policies relative to the ideal DP baseline, particularly under reduced observation windows?

3. Could the authors provide detailed wall-clock runtime and scaling plots comparing GNN inference and DP computation across a wider range of graph sizes and hardware setups?

4. Since the reward is binary (0/1, line 108), should the reward definition be refined or generalized for more nuanced pursuit outcomes?

5. How does Theorem 1 differ from the results in Lu et al., and is the proof in Appendix A.1 itself a novel contribution? Additionally, Equation (3) requires clearer explanation or motivation.

---

> ### Author Response · Authors · 2025-11-21
>
> Dear reviewer,
>
> Thank you very much for reviewing our paper and providing the constructive comments. Here we provide our response to the weaknesses and your questions.
>
> **Weakness 1**
>
> > The related work section is underdeveloped.
>
> Thank you for this comment, and we have added a related work section for the completeness of this paper. Due to the 10-page limitation, it is currently placed in Appendix E.
>
> **Weakness 2**
>
> > The test set is not particularly large or diverse, and key sensitivity analyses (e.g., observation range effects on trained RL policies) are missing.
>
> Thank you for pointing out the limitation of our evaluations. We have now created another set of test graphs with higher complexity from the seven famous locations in Table 1 (from Times Square to Sydney Opera House). Compared to the original graphs, the new graphs double both the map range and the discretization accuracy, leading to significantly larger node numbers (generally over $1000$). The results under these large graphs are reserved in our response to Question 3, showing no significant performance decline. In addition, we provide the results of how observation range affects the success rates of our RL policies. As is shown in the following table, the success rates increase with the observation range, which resembles the results of DP policies in the current Appendix D.2. This additional result implies that the RL policy trained under the observation range of $2$ can be directly applied to the cases with better sensing capabilities.
>
> | Observation Range | 2 | 3 | 4 | 5 | 6 |
> |---|---|---|---|---|---|
> | Grid Map | 1.00 | 1.00 | 1.00 | 1.00 | 1.00 |
> | Scotland-Yard Map | 0.76 | 0.98 | 0.99 | 0.99 | 1.00 |
> | Downtown Map | 0.99 | 0.99 | 1.00 | 1.00 | 1.00 |
> | Times Square, New York | 0.95 | 0.98 | 1.00 | 1.00 | 1.00 |
> | Hollywood Walk of Fame, LA | 0.38 | 0.59 | 0.96 | 1.00 | 1.00 |
> | Sagrada Familia, Barcelona | 0.20 | 0.72 | 0.88 | 0.95 | 0.96 |
> | The Bund, Shanghai | 0.25 | 0.55 | 0.82 | 0.82 | 0.83 |
> | Eiffel Tower, Paris | 1.00 | 1.00 | 1.00 | 1.00 | 1.00 |
> | Big Ben, London | 0.82 | 0.95 | 0.98 | 0.99 | 0.99 |
> | Sydney Opera House, Sydney | 0.95 | 0.98 | 1.00 | 1.00 | 1.00 |
>
> **Weakness 3**
>
> > The paper’s writing quality is uneven. Some abbreviations are undefined, and the structure is occasionally confusing (e.g., placing time complexity analysis within the evaluation section rather than methodology).
>
> Thanks for the comment on writing. We have now added clarifications for undefined abbreviations and restructured the paper. We move the time complexity analysis to the method section (Section 4.2) and place our additional experimental results in the evaluation section (Section 5.3).
>
> **Weakness 4**
>
> > Although the mathematical derivations for belief updates are technically sound, they would benefit from clearer illustrations or explanatory examples to improve accessibility for readers unfamiliar with partially observable multi-agent systems.
>
> Thank you for this comment. In our revised paper, we provide further explanations in the belief update section (Section 3.2) and complement our illustrations in Appendix B for better clarity.

---

> > ### Author Response · Authors · 2025-11-21
> >
> > **Question 1**
> >
> > > How does R2PS scale with an increasing number of pursuers or higher graph complexity (e.g., connectivity, degree, presence of cycles)? Can the theoretical guarantees extend to multi-pursuer settings as $m$ increases?
> >
> > Thanks for the question. The theoretical analysis in this paper holds for an arbitrary number of pursuers. Besides, the referenced paper [1] proposes a grouping mechanism for DP and a sequence model for EPG to guarantee that both of them can practically scale with an increasing number of pursuers. These techniques have been verified under at least $m=6$ pursuers in [1], and we have already employed the sequence model to represent multi-agent pursuer policies in R2PS. For the cases with more complicated graphs, we collectively show our additional results in the response to Question 3.
> >
> > **Question 2**
> >
> > > Is there a quantifiable bound on the sub-optimality of belief-averaged or RL-derived policies relative to the ideal DP baseline, particularly under reduced observation windows?
> >
> > Thanks for the question. While the DP pursuit strategy is provably worst-case optimal under full observability, there is not a quantifiable bound on the sub-optimality of the pursuit strategies under continual partial observability. We can consider a situation where the PEG graph is a tree with three long branches. In this case, one single DP pursuer with global information is sufficient to capture the evader by always approaching it. However, a pursuer with a fixed observation range far smaller than the branch length can never observe the evader in the worst case: as we know that the pursuer is essentially making a plan to explore the tree, we can always keep the evader lying in the branch different from the one that the pursuer is going to thoroughly explore. Then, the pursuer can never succeed as long as the evader is not initialized in the same branch.
> >
> > When the partial observability is not continual, however, the minimax policy (5) actually bounds the worst-case pursuit timesteps if the pursuers resume full observability after this step. As global information is rarely available in the real world, we further propose the belief-averaged policy (6) for a better empirical performance under continual partial observability.
> >
> > **Question 3**
> >
> > > Could the authors provide detailed wall-clock runtime and scaling plots comparing GNN inference and DP computation across a wider range of graph sizes and hardware setups?
> >
> > Thanks for the question, and here we provide our additional results.
> >
> > | Original Scale | Node Number | RL Success Rate | RL Inference Time (RTX A6000) | RL Inference Time (RTX 2080ti) | DP Inference Time (RTX 2080ti) |
> > |---|---|---|---|---|---|
> > | Times Square, New York | 171 | 0.95 | 0.007457 | 0.007686 | 0.2969 |
> > | Hollywood Walk of Fame, LA | 201 | 0.38 | 0.007496 | 0.007729 | 0.3523 |
> > | Sagrada Familia, Barcelona | 231 | 0.20 | 0.007565 | 0.007922 | 0.3860 |
> > | The Bund, Shanghai | 200 | 0.25 | 0.007486 | 0.007818 | 0.3521 |
> > | Eiffel Tower, Paris | 202 | 1.00 | 0.007524 | 0.007812 | 0.3646 |
> > | Big Ben, London | 192 | 0.82 | 0.007425 | 0.007674 | 0.3265 |
> > | Sydney Opera House, Sydney | 183 | 0.95 | 0.007477 | 0.007542 | 0.3332 |
> >
> > | Large Scale | Node Number | RL Success Rate | RL Inference Time (RTX A6000) | RL Inference Time (RTX 2080ti) | DP Inference Time (RTX 2080ti) |
> > |---|---|---|---|---|---|
> > | Times Square, New York | 1805 | 0.56 | 0.007563 | 0.009837 | 101 |
> > | Hollywood Walk of Fame, LA | 1251 | 0.46 | 0.007627 | 0.007917 | 33 |
> > | Sagrada Familia, Barcelona | 2065 | 0.33 | 0.008047 | 0.009895 | 139 |
> > | The Bund, Shanghai | 1723 | 0.46 | 0.007728 | 0.008117 | 83 |
> > | Eiffel Tower, Paris | 1825 | 0.41 | 0.007656 | 0.009616 | 96 |
> > | Big Ben, London | 1681 | 0.49 | 0.007530 | 0.007752 | 79 |
> > | Sydney Opera House, Sydney | 744 | 0.76 | 0.007523 | 0.007648 | 6 |
> >
> > The large-scale graphs are generated following the descriptions in our response to Weakness 2. As shown in the tables, the inference time of our RL policy is consistently below 0.01s under different graph sizes and hardware. However, the time of DP computations significantly increases with the graph sizes, even under CUDA speedups. We further provide the scaling plots in the current Appendix D.2. Besides, we can find that the overall performance of our RL pursuers does not suffer from a significant drop when facing large graphs, even though the policy is trained on relatively small graphs.

---

> > > ### Author Response · Authors · 2025-11-21
> > >
> > > **Question 4**
> > >
> > > > Since the reward is binary (0/1, line 108), should the reward definition be refined or generalized for more nuanced pursuit outcomes?
> > >
> > > Thanks for the question. This binary reward choice follows the common settings in the existing PEG research [1, 2, 3]. Considering more nuanced pursuit outcomes by refining the reward is possible, especially for RL policy learning. For example, the reward could be increased if the two pursuers capture the evader simultaneously. However, our DP-based theoretical results focus more on the minimum pursuit steps and are developed upon this simplest reward setting.
> > >
> > > **Question 5**
> > >
> > > > How does Theorem 1 differ from the results in Lu et al., and is the proof in Appendix A.1 itself a novel contribution? Additionally, Equation (3) requires clearer explanation or motivation.
> > >
> > > Thanks for the question. Theorem 1 is itself a result from Lu et al. [1], and its proof should not be considered as a novel contribution. The major theoretical contribution of this paper is Theorems 2 and 3, demonstrating that DP can induce strictly optimal policies under asynchronous moves. For Eq. (3), it differs from Eq. (2) in its input that describes asynchronous moves. The input allows for additional information $n_p$, which is the pursuers' choice of next positions in the current decision step. By observing this information in advance, the evader can decide based on the pursuers' positions after their decision rather than before. Therefore, we distinguish it from the setting of synchronous moves. Currently, we have added further explanations in Section 3.1.
> > >
> > > Thank you again for the insightful comments. We hope our response properly addresses your concerns. We are looking forward to having further discussions with you.
> > >
> > > **References**
> > >
> > > [1] Runyu Lu, Peng Zhang, Ruochuan Shi, Yuanheng Zhu, Dongbin Zhao, Yang Liu, Dong Wang, and Cesare Alippi. Equilibrium policy generalization: A reinforcement learning framework for cross-graph zero-shot generalization in pursuit-evasion games. In NeurIPS, 2025.
> > >
> > > [2] Pengdeng Li, Shuxin Li, Xinrun Wang, Jakub Cerny, Youzhi Zhang, Stephen McAleer, Hau Chan, and Bo An. Grasper: A generalist pursuer for pursuit-evasion problems. In AAMAS, 2024.
> > >
> > > [3] Shuxin Li, Xinrun Wang, Youzhi Zhang, Wanqi Xue, Jakub Cerny, and Bo An. Solving large-scale pursuit-evasion games using pre-trained strategies. In AAAI, 2023.

---

> > > ### Comment · Reviewer_N27S · 2025-11-25
> > >
> > > I appreciate the authors' efforts to resolve my concerns, and I find that almost all of my questions have been addressed.
> > >
> > > However, authors still do not provide the results about a larger $m$ instead of 2. As for computation cost, could you consider FLOPs which is independent with GPU?

---

> ### Author Response · Authors · 2025-11-26
>
> Thank you for the comments and additional suggestions.
>
> Currently, we are conducting experiments in the cases of $m=4$ and $m=6$, based on the grouping mechanism established by Lu et al. We will show the additional results upon finishing. Besides, since the DP algorithm is completely based on integer operations (with no floating-point operations), we are considering providing the FLOPs comparison of our RL inference under different graph sizes and agent numbers.

---

> ### Author Response · Authors · 2025-11-28
>
> Currently, we have finished our further experiments under larger pursuer numbers. The following table compares the pursuit success rates of the multi-agent RL policies against the asynchronous-move DP evader under different pursuer numbers $m$. Clearly, the pursuer policy under $m=4$ guarantees success rates significantly higher than the original ones under $m=2$. When $m=6$, the success rates are close to $1$ even under the fixed observation range of $2$.
>
> | Pursuer Number | $m=2$ | $m=4$ | $m=6$ |
> |--------------|-----|-----|-----|
> | Grid Map | 1.00 | 1.00 | 1.00 |
> | Scotland-Yard Map | 0.76 | 0.99 | 1.00 |
> | Downtown Map | 0.99 | 1.00 | 1.00 |
> | Times Square, New York | 0.95 | 0.97 | 1.00 |
> | Hollywood Walk of Fame, LA | 0.38 | 0.82 | 0.93 |
> | Sagrada Familia, Barcelona | 0.20 | 0.74 | 0.94 |
> | The Bund, Shanghai | 0.25 | 0.99 | 1.00 |
> | Eiffel Tower, Paris | 1.00 | 1.00 | 1.00 |
> | Big Ben, London | 0.82 | 1.00 | 1.00 |
> | Sydney Opera House, Sydney | 0.95 | 0.99 | 1.00 |
>
> Additionally, we use Figure 7 in the current Appendix D.3 to further illustrate the FLOPs of our RL inference. As is shown in the figure, the actual FLOPs increase in a near-linear manner with respect to both graph size and pursuer number. This result further supports the scalability of our R2PS approach.

---

### Author Response · Authors · 2025-11-21
**Response to Reviewers**

Dear reviewers,

Thank you for reviewing our paper. We greatly appreciate your effort in reading the paper and providing valuable comments. Based on your comments and suggestions, we have made a revision to our manuscript and updated the submission. The additional tables and figures have been marked in blue, including:

1. Table 3 (RL Success Rate and Comparison of Inference Time in Large Graphs) [Section 5.3]

2. Table 4 (RL Success Rate Comparison under Different Belief Update Conditions) [Section 5.3]

3. Figures 3&4 (Pursuit Illustration under Belief Preservation) [Appendix B]

4. Table 7 (Success Rates of RL Pursuers under Different Observation Ranges) [Appendix D.2]

5. Figure 6 (Scaling Plots of RL and DP Inference Time) [Appendix D.2]

We will be happy if our responses and modifications can address your concerns and improve your appreciation of this work. If you find anything unclear, please feel free to point it out. We are more than delighted to have further discussions and improve our manuscript.

---

> ### Author Response · Authors · 2025-11-28
>
> Thanks to the additional suggestions from Reviewer N27S, we include a new subsection (Appendix D.3) to further demonstrate the scalability of our approach with respect to pursuer number $m$. The new results are listed as follows:
>
> 6. Table 8 (Success Rates of RL Policy under Different Pursuer Numbers) [Appendix D.3]
>
> 7. Figure 7 (Scaling Plots of Floating-Point Operations under Different RL Pursuer Numbers) [Appendix D.3]
>
> We are delighted to see that our responses have been acknowledged by all reviewers, addressing the major concerns regarding this paper. We would like to thank all reviewers once again for your great effort and invaluable comments!

---

### Meta-Review · Area_Chair_gzWW · 2026-01-06

**Summary:**

The paper presents a well-executed and timely contribution to pursuit–evasion games by extending dynamic programming guarantees to asynchronous moves and partial observability, and by integrating a principled belief-preservation mechanism into cross-graph reinforcement learning to yield real-time, worst-case robust pursuer policies. While some reviewers initially raised concerns regarding novelty relative to prior EPG work, scalability, and experimental coverage, the authors provided thorough rebuttals with substantial new analyses (larger graphs, additional ablations, observation-range and pursuer-count scaling, runtime/FLOPs) that satisfactorily addressed these points. Multiple reviewers expressed satisfaction after the revisions and maintained or increased their scores, noting the solid theoretical grounding, clear practical relevance, and strong empirical validation, including zero-shot generalization on real-world graphs with real-time inference advantages over DP and PSRO. Given the overall positive reception and the strengthened final submission, I recommend acceptance.

**Reviewer Concerns:**

Most reviewer concerns were addressed.

**Reviewer Scores:**

Most will change their score positively

---

### Decision · Program_Chairs · 2026-01-26

Accept (Poster)